# Cell-sized confinement controls generation and stability of a protein wave for spatiotemporal regulation in cells

Shunshi Kohyama[1], Natsuhiko Yoshinaga[2,3]*, Miho Yanagisawa[4], Kei Fujiwara[1]*, Nobuhide Doi[1]

[1]Department of Biosciences and Informatics, Keio University, Yokohama, Japan; [2]Mathematical Science Group, WPI Advanced Institute for Materials Research (WPI-AIMR), Tohoku University Katahira, Sendai, Japan; [3]MathAM-OIL, AIST, Sendai, Japan; [4]Department of Basic Science, Komaba Institute for Science, Graduate School of Arts and Sciences, The University of Tokyo, Tokyo, Japan

**Abstract** The Min system, a system that determines the bacterial cell division plane, uses changes in the localization of proteins (a Min wave) that emerges by reaction-diffusion coupling. Although previous studies have shown that space sizes and boundaries modulate the shape and speed of Min waves, their effects on wave emergence were still elusive. Here, by using a microsized fully confined space to mimic live cells, we revealed that confinement changes the conditions for the emergence of Min waves. In the microsized space, an increased surface-to-volume ratio changed the localization efficiency of proteins on membranes, and therefore, suppression of the localization change was necessary for the stable generation of Min waves. Furthermore, we showed that the cell-sized space strictly limits parameters for wave emergence because confinement inhibits both the instability and excitability of the system. These results show that confinement of reaction-diffusion systems has the potential to control spatiotemporal patterns in live cells.

DOI: https://doi.org/10.7554/eLife.44591.001

*For correspondence:
yoshinaga@tohoku.ac.jp (NY);
fujiwara@bio.keio.ac.jp (KF)

**Competing interests:** The authors declare that no competing interests exist.

## Introduction

Spatiotemporal self-organization of biomolecules in cells is part of a fundamental mechanism to maintain cellular structure. In particular, the intracellular reaction-diffusion wave (iRD) is an essential mechanism for various processes of spatiotemporal regulation, including DNA segregation (*Adachi et al., 2006*), cell-shape deformation, cell migration (*Arai et al., 2010*; *Huang et al., 2013*), and cell polarization (*Goryachev and Pokhilko, 2008*). A remarkable example of iRD is Min wave, which is a bacterial spatiotemporal organization system (Min system). The Min system places the division site precisely at the center of the cell by using iRD (*Rothfield et al., 2005*; *Rowlett and Margolin, 2013*). This system comprises three proteins called MinC, MinD, and MinE, with the localization of MinD and MinE oscillating between one pole and the other as the result of a coupling between biochemical reactions and molecular diffusions (*Loose et al., 2008*; *Rowlett and Margolin, 2013*). MinC has no role in the Min wave but rather inhibits the polymerization of a cell division initiation factor (FtsZ) by following the Min wave. This process enforces the initiation of cell division only at the center of cells (*Rothfield et al., 2005*; *Rowlett and Margolin, 2013*).

To date, Min wave is the only biological RD system reconstituted in vitro. The reconstitution of Min wave was first shown by spotting a mixture of MinD, MinE, and ATP on two-dimensional (2D) planar membranes comprising *E. coli* polar lipid extract in open geometry (*Loose et al., 2008*). The following studies based on a 2D planar system have clarified the mechanisms of wave generation and

the characteristics of Min waves (*Loose et al., 2011*; *Martos et al., 2013*; *Zieske and Schwille, 2013*; *Vecchiarelli et al., 2014*; *Zieske et al., 2016*; *Caspi and Dekker, 2016*). In vitro studies have demonstrated that external environments such as boundary shapes, protein concentrations, and lipid species alter the patterns, velocities, wavelengths and shapes of Min waves (*Martos et al., 2013*; *Zieske and Schwille, 2013*; *Vecchiarelli et al., 2014*; *Zieske et al., 2016*; *Caspi and Dekker, 2016*; *Denk et al., 2018*). In particular, boundary shapes prepared by structured poly(dimethylsilox-ane) (PDMS) chambers significantly change the behavior of Min waves, with studies showing that a rod-shape is important in terms of inducing the pole-to-pole oscillation found in living cells (*Zieske and Schwille, 2013*; *Zieske et al., 2016*; *Caspi and Dekker, 2016*).

Owing to the importance of Min waves in initiating division at a precise location, the timing, conditions, and regulation of their emergence should be investigated. The critical conditions for Min wave emergence, including environmental effects, have been surveyed in open spaces, but the effect of confinement in cell-sized spaces, which is one of the most remarkable features of living cells, has been poorly addressed. Although some studies have reported reconstitution of Min waves in fully confined cell-sized spaces (*Zieske et al., 2016*; *Caspi and Dekker, 2016*; *Litschel et al., 2018*), lipid conditions were modified or the spaces were closed after observing wave generation. The necessity of these treatments suggests that the cell-sized space affects the conditions that are needed for Min wave emergence.

Recent studies have unveiled that confinement inside cell-sized space alters both the behaviors of biochemical reactions and molecular diffusion (*Yanagisawa et al., 2014b*; *Küchler et al., 2016*; *Watanabe and Yanagisawa, 2018*). Because RD waves appear only in limited parameter ranges (*Zhabotinsky et al., 1995*; *Epstein and Showalter, 1996*), encapsulation inside a cell-sized space should shift the conditions that are suitable for Min wave emergence, such as the diffusion and interaction of the wave's elements. Moreover, by considering interference of the RD waves at the time of two-wave collision (*Lee et al., 1994*) and the initiation of Min waves by interactions among Min proteins on 2D planar membranes (*Loose et al., 2008*), it is plausible that the conditions for the emergence of a single wave in a small space are different from those that support multiple waves in a large space.

In this study, we investigated the mechanism behind the generation of Min waves in a closed micro-sized space that was fully covered with *Escherichia coli* polar lipid extract. Our experimental and theoretical analyses revealed that a fully confined micro-sized space changes the rates of protein localization, and therefore, that elements that cancel this effect are necessary to produce Min waves in a small space. Furthermore, our results show that the cell-sized space itself plays some role in spatio-temporal regulation via RD mechanisms in living cells.

## Results

### MinDE are insufficient for emerging Min waves in a micro-sized space fully covered with *E. coli* polar lipid extract

Previous studies have reported that only MinD, MinE, and ATP are necessary and sufficient for the generation of Min waves on 2D planar membranes of *E. coli* polar lipid extract (*Loose et al., 2008*) (*Figure 1A*) and in a cell-sized space that is fully confined by using a modified lipid mixture (*Zieske et al., 2016*; *Litschel et al., 2018*). The Min wave in open geometry has been well characterized in many laboratories (*Loose et al., 2011*; *Vecchiarelli et al., 2014*; *Caspi and Dekker, 2016*), and has also been reproduced by using materials prepared in our laboratory (sfGFP-MinD and MinE-mCherry) (*Figure 1B*). We encapsulated these materials in micro-sized spaces that were fully covered with *E. coli* polar lipid extract using an emulsification method (*Fujiwara and Yanagi-sawa, 2014*; *Zieske et al., 2016*). However, we found that sfGFP-MinD, MinE-mCherry, and ATP are insufficient for Min wave emergence in the microdroplets covered with *E. coli* polar lipids (*Figure 1C*, *Video 1*). Use of non-fluorescent tagged MinDE tracked by sfGFP-MinC indicated that fluorescent proteins fused with MinD or MinE could not explain the lack of wave occurrence (*Figure 1A,B*). By contrast, sfGFP-MinD, MinE-mCherry, and ATP induced Min waves in microdroplets covered with a lipid mixture (85% DOPC and 15% cardiolipin), as reported previously (*Zieske et al., 2016*) (*Video 2*). These results indicated that after encapsulation in a micro-sized

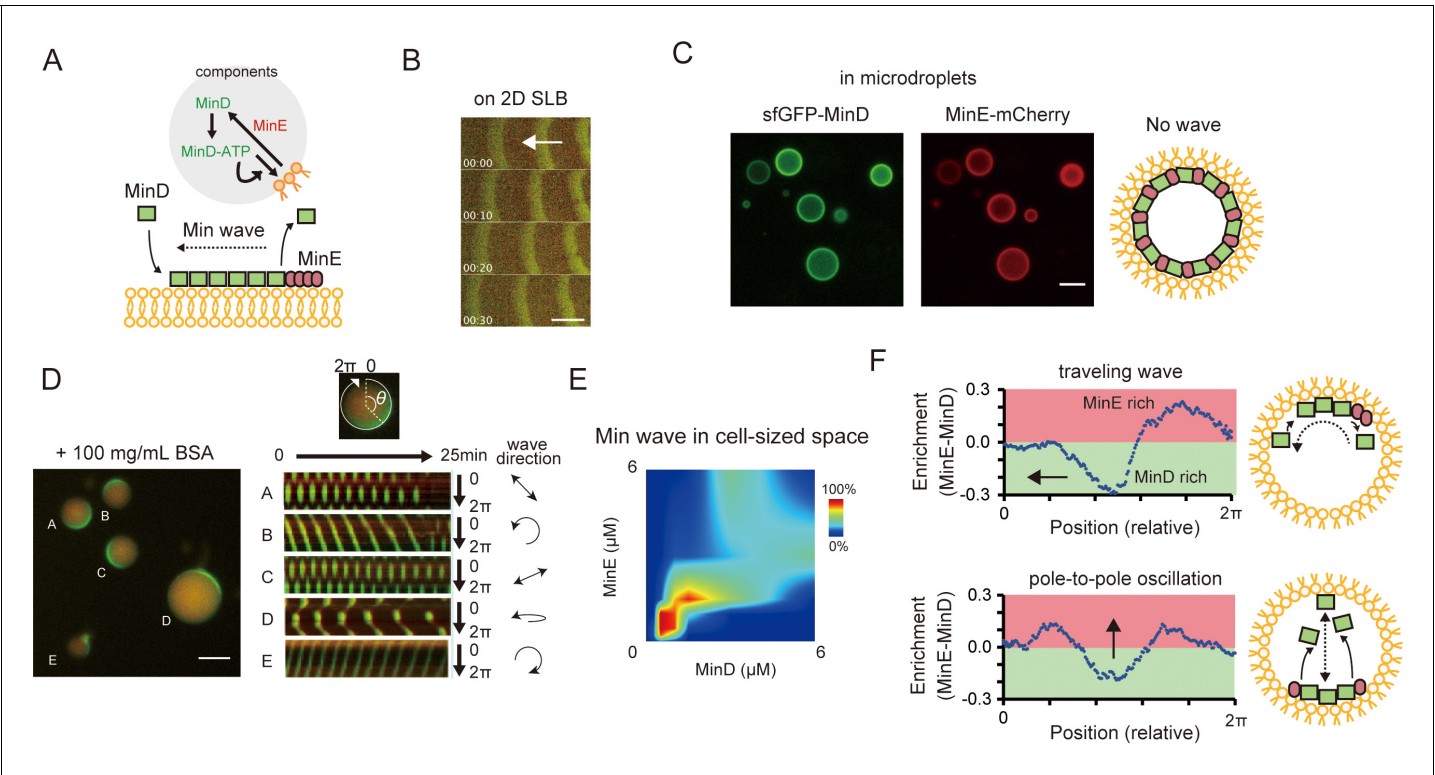

**Figure 1.** Min waves emergence in microdroplets as a result of high-concentration BSA addition. (**A**) Schematic illustration of a simplified molecular mechanism underlying Min wave propagation and the experimental system. (**B**) Min waves on 2D supported lipid bilayers (SLB). (**C,D**) Microdroplets encapsulating 1 μM sfGFP-MinD, MinE-mCherry, and 2.5 mM ATP in the absence (**C**) or the presence of 100 mg/mL BSA (**D**). Scale bars: 10 μm. (**D**) Kymographs of sfGFP-MinD (green) and MinE-mCherry (red) in the proximity of membranes in each droplet are shown in the right half of the panel. Kymographs were generated by tracking fluorescence intensities along circumference lines on the membrane surface. Arrows beside the kymographs show the direction and mode of the Min wave. Single-round and double-headed arrows indicate a traveling wave and pole-to-pole oscillation, respectively. (**E**) Probability of inhomogeneous localization and wave propagation revealed by the reconstitution experiments at various concentrations of sfGFP-MinD and MinE-mCherry in microdroplets. (**F**) Enrichment profiles of MinD and MinE derived from normalized surface plots.

DOI: https://doi.org/10.7554/eLife.44591.002

The following source data and figure supplements are available for figure 1:

**Source data 1.** Numerical data of *Figure 1E*.
DOI: https://doi.org/10.7554/eLife.44591.006
**Figure supplement 1.** Tracking MinD by fluorescence-tagged MinC.
DOI: https://doi.org/10.7554/eLife.44591.003
**Figure supplement 2.** ATP dependence of the Min system for wave propagation in microdroplets containing 100 mg/mL BSA.
DOI: https://doi.org/10.7554/eLife.44591.004
**Figure supplement 3.** Time-lapse images of propagation waves in microdroplets.
DOI: https://doi.org/10.7554/eLife.44591.005

space fully covered with lipid alters, some critical parameters for Min wave emergence differ from those that allow for wave emergence on 2D membranes.

## Addition of protein crowder assists Min wave emergence in cell-sized droplets

The difference in emergence conditions between a 2D planar membrane and a 3D closed space raised the possibility that factors other than MinDE are involved in the regulation of Min wave emergence in living cells. From the fact that lipid species change the conditions for Min wave emergence, we assumed that changes in the balance between reaction and diffusion by factors such as the physicochemical environments are associated with this difference.

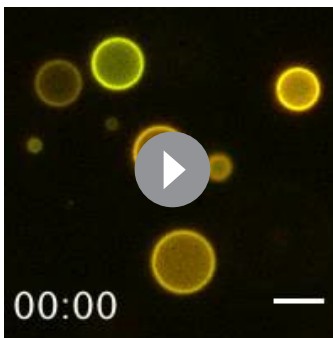

**Video 1.** Behaviors of Min proteins entrapped in microdroplets.
DOI: https://doi.org/10.7554/eLife.44591.007

As a candidate for such a factor, we focused on molecular crowding in cells. In living cells, ~30% of the cell mass consists of macromolecules, and such crowding of molecules modulates both biochemical reactions and molecular diffusion (*Zhou et al., 2008*; *Groen et al., 2015*). Therefore, crowding is probably associated with patterns that affect reaction-diffusion systems. In fact, crowding agents that emulate molecular crowding in vitro have been shown to affect the coupling of Min waves over membrane gaps (*Schweizer et al., 2012*) and the wavelength of Min waves in the presence of the FtsZ system (*Martos et al., 2015*).

To test this possibility, synthetic polymers (PEG8000 or Ficoll70) or a protein-based crowding agent (BSA) were mixed with MinDE and ATP, with the mixtures then being encapsulated in microdroplets covered with *E. coli* polar lipid. Remarkably, co-supplementation of BSA at high concentration (100 mg/mL) with Min proteins induced Min wave emergence (*Figure 1D*, *Video 3*), whiereas neither PEG8000 nor Ficoll70 induced Min waves (*Videos 4* and *5*). Supplementation of BSA also induced Min-wave emergence when no-fluorescence-tagged MinDE was tracked by sfGFP-MinC (*Figure 1—figure supplement 1C*, *Video 6*).

Varying concentrations of MinD and MinE indicated that both proteins should be present at a concentration of around 1 µM to lead the emergence of Min waves (*Figure 1E*), consistent with their concentrations in vivo (*Shih et al., 2002*). ATP replacement with ADP or ATPγS, or replacement of MinD with an ATPase-deficient mutant (*Zhou et al., 2005*), showed that the Min wave depends on ATP (*Figure 1—figure supplement 2*), as is the case for waves on 2D planar membranes (*Loose et al., 2008*). The frequent patterns observed were pole-to-pole oscillations and traveling waves (*Figure 1—figure supplement 3*), as noted by a previous study using modified lipids (*Zieske et al., 2016*). MinE was enriched at the tail of the traveling wave (*Figure 1F* top) and was enriched at both tails of the wave in pole-to-pole oscillations (*Figure 1F* bottom). These MinE enrichments were similar to those reported previously for traveling waves on 2D planar membranes (*Loose et al., 2008*), and for the so-called E ring observed in living cells with pole-to-pole oscillations (*Rothfield et al., 2005*).

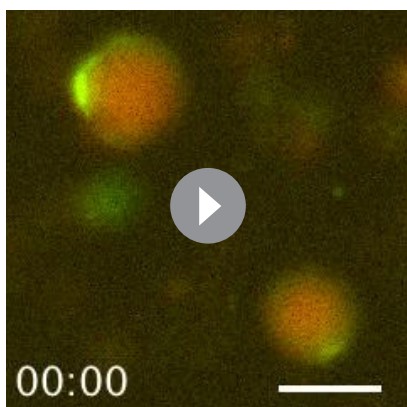

**Video 2.** Wave propagation of Min proteins in microdroplets with a lipid mixture (85% DOPC and 15% cardiolipin).
DOI: https://doi.org/10.7554/eLife.44591.008

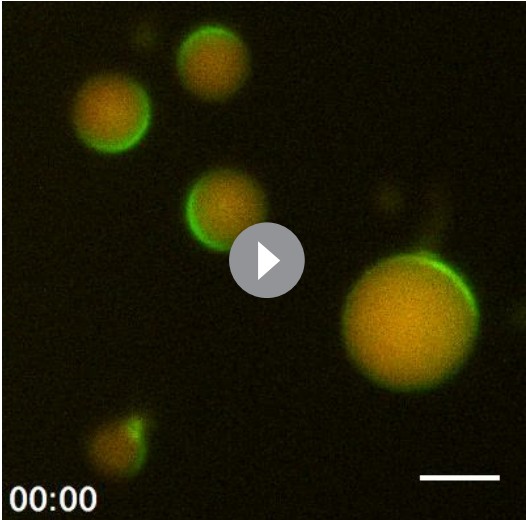

**Video 3.** Wave propagation of Min proteins in microdroplets containing 100 mg/mL BSA.
DOI: https://doi.org/10.7554/eLife.44591.009

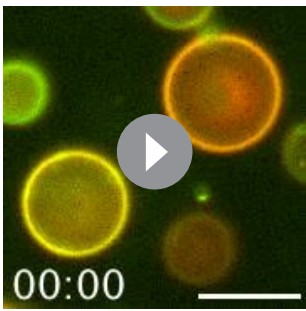

**Video 4.** Behavior of Min proteins entrapped in microdroplets containing 100 mg/mL PEG8000.
DOI: https://doi.org/10.7554/eLife.44591.010

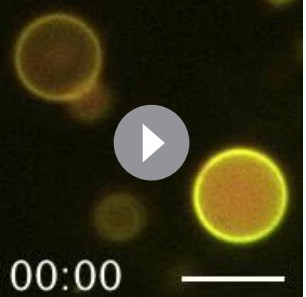

**Video 5.** Behavior of Min proteins entrapped in microdroplets containing 100 mg/mL Ficoll70.
DOI: https://doi.org/10.7554/eLife.44591.011

## BSA modifies the attachment of MinE onto membranes without MinD

To understand why the conditions for the emergence of Min waves differ between 2D planar membranes and 3D closed geometry, we investigated how BSA affects the mechanism of wave emergence in a closed micro-sized space. Crowding agents such as BSA may change reaction rates and diffusion rates (*Zhou et al., 2008*; *Groen et al., 2015*). Changes to the reaction rate are related to changes in the interactions between Min proteins or between a protein and a membrane. The first-known mechanism to modify these interactions is the depletion force, which enhances the attraction between proteins. The crowding agents might also bind directly with MinD or MinE to promote the formation of MinDE complexes. Furthermore, the crowding agents decrease the diffusion constants. We investigated these effects in detail. Among these effects, the effect of depletion force was excluded from the investigation because previous studies have indicated that BSA causes a much weaker depletion force than PEG8000 or Ficoll70 (*Groen et al., 2015*).

The diffusion of macromolecules inside the closed space was evaluated using fluorescence correlation spectroscopy (FCS) and fluorescence recovery after photo-bleaching (FRAP). FCS revealed that the diffusion rate of GFP in cytosolic parts at 50 mg/mL BSA was similar to that in non-crowding conditions but decreased at over 100 mg/mL (*Figure 2A*). However, we found that Min waves were generated stably even with 50 mg/mL of BSA (*Figure 2—figure supplement 1*). The effect of BSA on the diffusion of sfGFP-MinD on membranes was investigated using FRAP. The diffusion rates of MinD on lipid membranes of various sizes of microdroplets decreased only slightly even in 100 mg/mL of BSA (*Figure 2B*).

To test how BSA affects the reactions that form MinDE complexes, we employed a pull-down assay to analyze the direct association of BSA with MinD or MinE. BSA was mixed with MinD or MinE immobilized on Ni-NTA beads using a histidine-tag. The pull-down assay showed that BSA flowed through the Ni-NTA with Min proteins, and therefore, no BSA band was found after eluting MinD or MinE by imidazole. These results indicate that BSA does not bind directly to MinD or MinE. The pull-down assay using MinD$^{D40A}$Δ10, which still bound with MinE due to its lack of ATPase activity (*Park et al., 2017*), also indicates that BSA does not enhance the interactions of the MinDE complex (*Figure 2C*).

Even though the three above-mentioned effects are not relevant, we found that BSA leads MinE to localize differently in cytosolic parts and on membranes. Each sfGFP-MinD and MinE-mCherry was encapsulated in a micro-sized

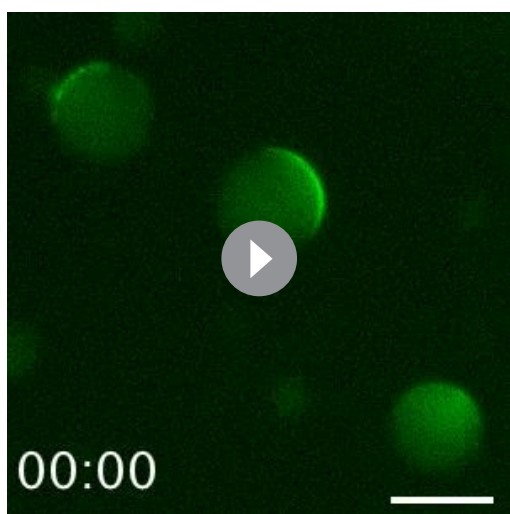

**Video 6.** Wave propagation of non-tagged MinD tracked by sfGFP-MinC in microdroplets containing 100 mg/mL BSA.
DOI: https://doi.org/10.7554/eLife.44591.012

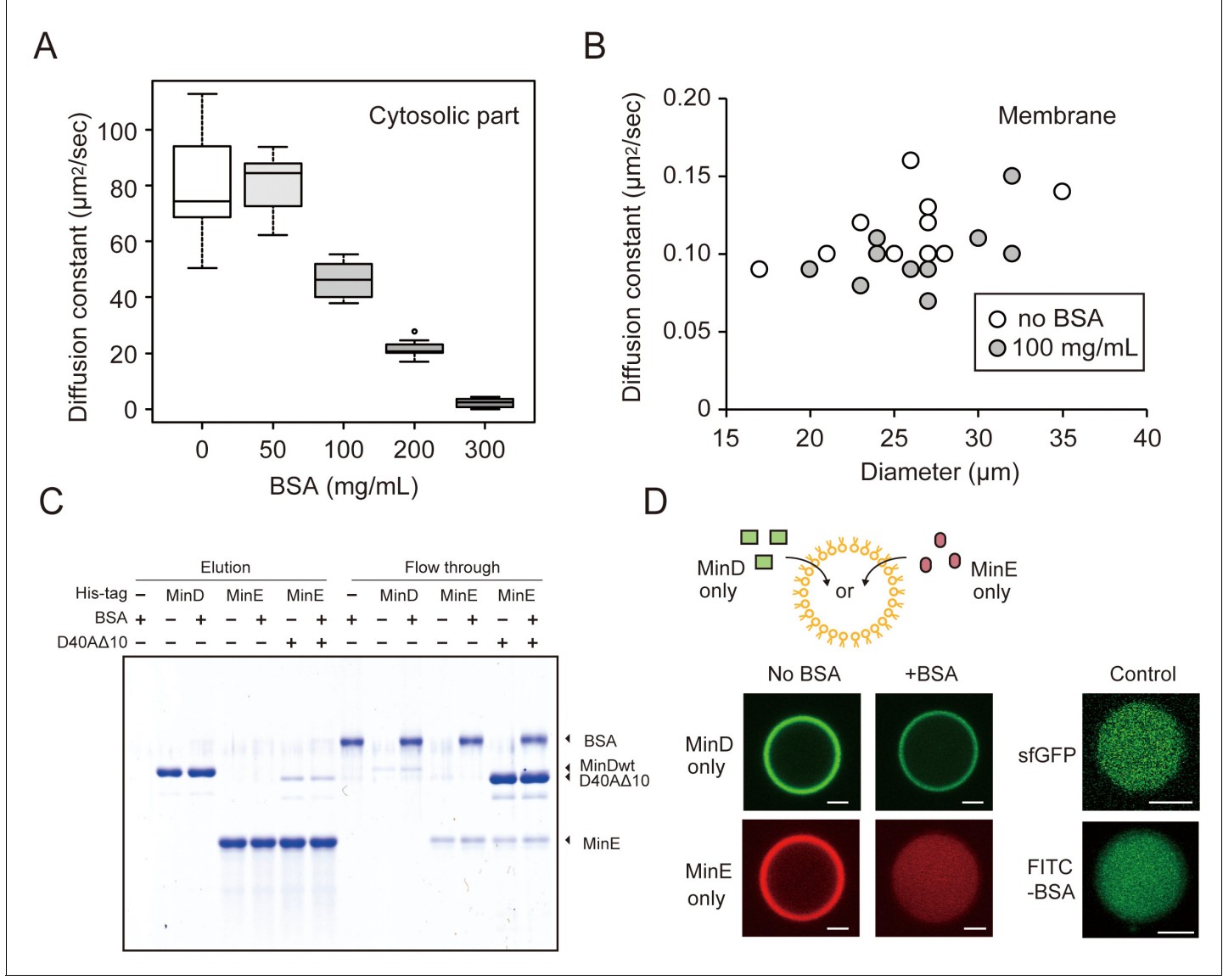

**Figure 2.** The effects of BSA on Min wave elements in cell-sized droplets. (**A**) Diffusion coefficients of sfGFP in solutions with various BSA concentrations entrapped in microdroplets (n = 10). (**B**) Diffusion coefficients of sfGFP-MinD attached on lipid membranes with or without 100 mg/mL BSA (n = 10) plotted as a function of microdroplet diameters. (**C**) Pull-down assay for BSA and Min proteins. His-sfGFP-MinD, MinE-mCherry-His, or sfGFP-MinD$^{D40A}$Δ10 with MinE-mCherry-His was incubated with Ni-NTA resins with or without BSA. The fractions eluted by imidazole and the flow through were visualized by CBB staining. (**D**) Inhibition of spontaneous binding between membranes and Min proteins by BSA. Either sfGFP-MinD or MinE-mCherry was encapsulated in the presence or absence of 50 mg/mL BSA. Microdroplets of 20 μm diameter are shown. As a control, the same experiments were performed using 1 μM sfGFP only or 5 μM FITC-labelled BSA with 5 μM BSA (total 0.67 mg/mL). Scale bars represent 5 μm.

DOI: https://doi.org/10.7554/eLife.44591.013

The following figure supplements are available for figure 2:

**Figure supplement 1.** Traveling waves emerging in microdroplets containing 50 mg/mL BSA.
DOI: https://doi.org/10.7554/eLife.44591.014

**Figure supplement 2.** Diffusion coefficients of MinE-mCherry attached on lipid membranes without BSA.
DOI: https://doi.org/10.7554/eLife.44591.015

space that was fully covered with *E. coli* polar lipid, and the localization of MinD and MinE was visualized using a confocal fluorescence microscope. In the absence of BSA, almost all of the MinD and MinE were localized similarly on membranes. By contrast, the addition of BSA drastically changed the localization. In the presence of BSA, changes in MinD localization were relatively few, but

the localization of MinE on membranes completely disappeared (*Figure 2D*). Because sfGFP alone or BSA at low concentration (0.67 mg/mL) does not localize on membranes (*Figure 2D*), the spontaneous localization of MinE cannot be explained by membrane defects. Furthermore, FRAP also showed that MinE-mCherry diffuses faster than sfGFP-MinD in the absence of BSA, which indicates that the localization of MinE was not driven by protein denaturation (*Figure 2—figure supplement 2*). These results suggest that changes in the localization of MinE are a key factor in the emergence of waves in microdroplets.

## Suppression of spontaneous membrane localization of MinE is the key to emergence of Min waves in micro-sized space

To quantify the details of MinE localization in microdroplets, we employed an index value for the localization ratio (c/m) obtained by dividing the concentration of MinE in the cytosolic parts (c [1/µm$^3$]) by those on membranes (m [1/µm$^2$]) (*Figure 3A*). Both concentrations are expressed by characteristic concentrations in the cytosol, $c_b$, and on the membrane, $c_s$, such as $\mathrm{c} = c_0 c_b$ and $\mathrm{m} = m_0 c_s$, respectively. Here, we may freely choose the values of the characteristic concentrations, which would accordingly change the values of the unitless concentrations $c_0$ and $m_0$. It is reasonable to assume that these quantities are proportional to the fluorescence intensity at the position in the cytosol $I_b$ and on the membrane $I_s$, possibly with different proportional constants such as $c_0 = \alpha_b I_b$ and $m_0 = \alpha_s I_s$, respectively. This argument ensures that the localization ratio (c/m) is identified as $\mathrm{c/m} = (\alpha_b c_b/(\alpha_s c_s))I_b/I_s$ with the ratio of fluorescence intensity up to a proportional constant. This argument implies that the relative value of c/m is a relevant quantity.

We measured fluorescence intensities at the center of microdroplets and the edges of signals. In this case, c/m becomes one when MinE is not localized on the membrane, whereas c/m becomes 0 when all MinE localizes on membranes. As shown in *Figure 3B*, the c/m of MinE increased in proportion to BSA concentration.

Then, we investigated the relation between c/m and the probability of Min wave emergence. Plots of wave emergence percentage as a function of c/m controlled by BSA concentration showed that its relation is a sigmoidal as a threshold function (*Figure 3C*). Min waves were observed in a small fraction of microdroplets at c/m <0.4 (<1 mg/mL BSA), and in almost all microdroplets at c/m >0.7 (>30 mg/mL BSA).

To check whether or not the effect is specific to BSA, we tested another protein crowder — a cell extract of *E. coli* prepared by sonication (*Groen et al., 2015*). In this case, we added an ATP recycling system to suppress ATP deletion caused by the components of the cell extract. The cell extract modulated the c/m of MinE in a similar manner as BSA, although its effect was stronger than BSA (*Figure 3D*). Moreover, the cell extract also led to the emergence of Min waves in the microdroplets covered with *E. coli* polar lipid extract (*Video 7*). The relation between c/m and the probability of Min wave emergence was similar to that of BSA concentration (*Figure 3E*). These results indicated that high c/m is required for the stable emergence of Min waves in a 3D closed geometry.

Under conditions using macromolecular crowding reagents that do not lead to the emergence of Min waves (PEG8000 and Ficoll70), c/m was as low as and similar to that without BSA (*Figure 3F*). Then, we checked c/m in the case of microdroplets covered with the modified lipid (15% cardiolipin and 85% DOPC), which causes Min waves without BSA. When the lipid was modified, c/m was near 0.4 (*Figure 3F*), which is as high as the minimal BSA-associated c/m value required for Min wave emergence. These results supported the notion that the suppression of attachment of MinE on membrane without the aid of MinD is key to the emergence of Min waves in micro-sized space.

## Smaller microdroplets have the higher rate of spontaneous MinE membrane localization

To determine the c/m of MinE, the maximum levels of attachment on membranes and the total amounts of MinE are conceivable factors. In smaller microdroplets, the surface-area-to-volume ratio is large, and therefore, almost all MinE can localize on the membrane (meaning c/m ~0). By contrast, in large spaces, the levels of MinE on membranes are close to the maximum levels for membrane localization (such as those found on 2D lipid bilayers) and not all of the MinE can localize at the membranes; this leads to a larger c/m. In fact, c/m was estimated to be 0.76 in the case of 2D lipids that showed Min wave emergence in the absence of BSA (*Figure 4—figure supplement 1*). If this

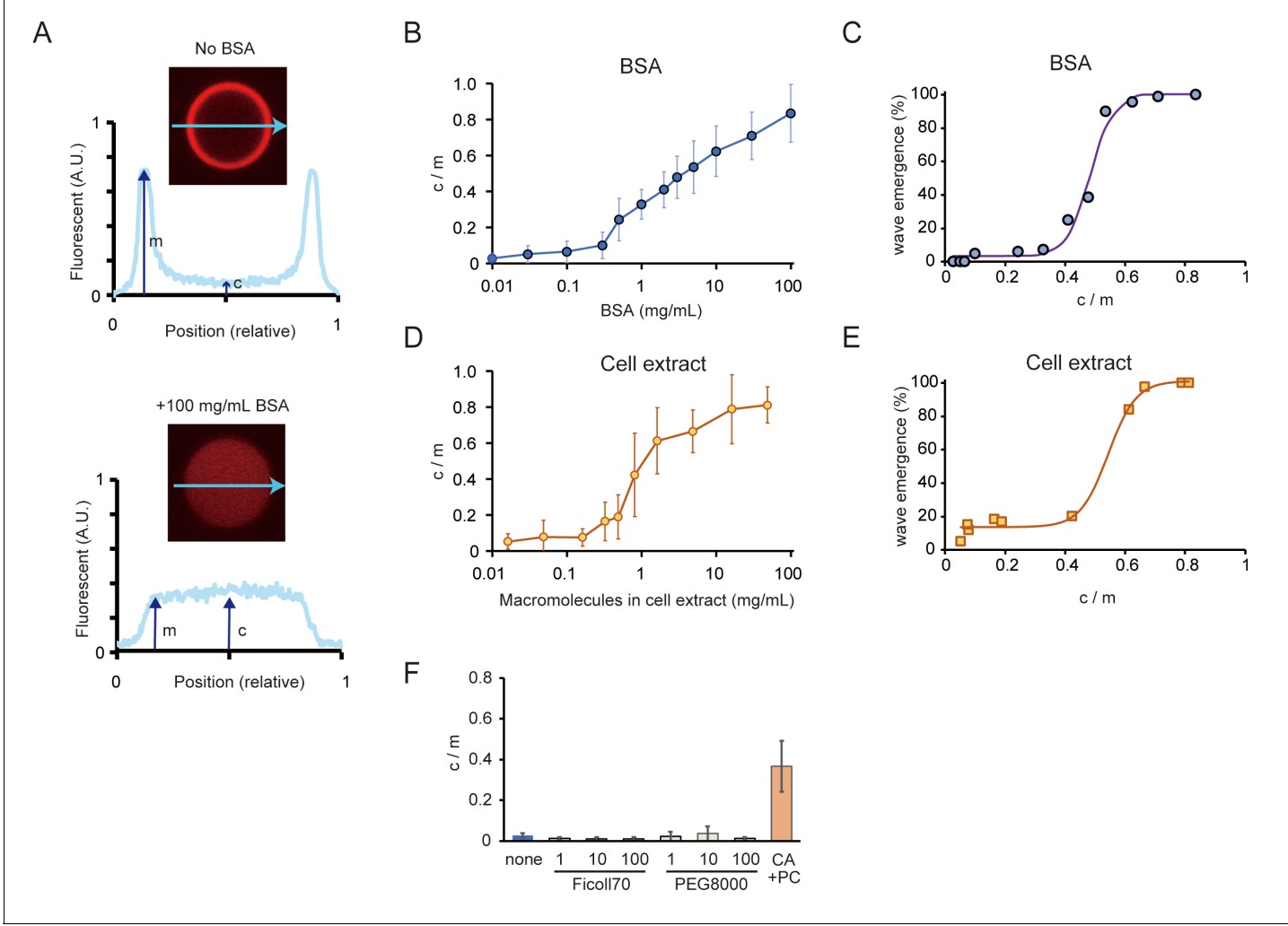

**Figure 3.** Relations between the rate of spontaneous localization of MinE on membranes and Min wave emergence. (**A**) Schematic illustrations of the method to evaluate MinE localization (c/m). (**B, D**) Changes of c/m of MinE-mCherry at various BSA concentrations (**B**) and for different concentrations of macromolecules in the cell extract (**D**). (**C, E**) Percentages of microdroplets that have a Min wave plotted as a function of c/m as determined by BSA concentration(**B**) or concentration of *E. coli* cell extract (**D**), respectively. The fitting lines are sigmoidal curves. (**F**) Effects of macromolecular crowding agents (1, 10, 100 mg/mL) and the modified lipid condition (15% cardiolipin and 85% DOPC condition, abbreviated as CA + PC) on the c/m of MinE-mCherry. Microdroplets smaller than 30 μm in diameter were selected and the c/m of 1 μM MinE-mCherry was evaluated.

DOI: https://doi.org/10.7554/eLife.44591.016

assumption holds, c/m will increase in proportion to the sizes of the microdroplets, and its response to the space size is therefore sensitive to the concentration of MinE used.

To verify this point, we investigated the localization of MinE in various sizes of microdroplets in the absence of BSA. In smaller microdroplets (<20 μm diameter), c/m was less than 0.2. In larger microdroplets (≥20 μm diameter), c/m increased in proportion to the amount of MinE. Moreover, the increase in c/m was highly dependent on the concentration of MinE. The diameters of microdroplets in which c/m reached 0.5 were approximately 45 μm at 10 μM MinE, and around 70 μm at 3 μM MinE (*Figure 4*). In the case of 1 μM MinE, c/m stayed low when the droplet size was less than 130 μm (*Figure 4*). We also tested the size dependence of c/m in the presence of 10 mg/mL BSA, which is the minimum BSA concentration for Min wave emergence in microdroplets. In this case, c/m did not depend on microdroplet size. However, the value of c/m was higher than 0.7 (*Figure 4—figure supplement 2*).

Although c/m is higher than 0.4 in >50 μm microdroplets entrapping 3 μM MinE (*Figure 4*), no Min waves were observed at 1 μM MinD and 3 μM MinE in the absence of BSA. This result may be

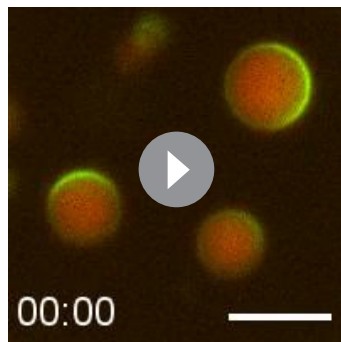

**Video 7.** Wave propagation of Min proteins in microdroplets containing 16 mg/mL macromolecules in cell extract.
DOI: https://doi.org/10.7554/eLife.44591.017

associated with the very low rate of Min wave appearance at this MinE concentration, even in the presence of 100 mg/mL BSA (**Figure 1E**).

## Computational simulation of a Min wave supports the importance of MinE localization for wave emergence

To understand the importance of MinE localization, we examined Min wave generation using computational simulations (see 'Materials and methods' and 'Appendices 1–3'). We considered two models. Model I (**Figure 5—figure supplement 1A**) is simply based on the model proposed by **Bonny et al. (2013)**, whereas Model II (**Figure 5—figure supplement 1B**) is based on a combination of the two models proposed by **Bonny et al. (2013)** and by **Huang et al. (2003)** in order to incorporate the effects of persistent MinE membrane binding and transformation from ADP-MinD to ATP-MinD in the cytosol (see Appendix 1). On the basis of these models, we investigated the effect of spontaneous MinE binding. This effect is characterized by the quantity $c_{e,0}$, which demonstrates the concentration of MinE on the membrane in the absence of MinD. To our knowledge, all the previous models lack this effect, that is, MinE was assumed to be in the cytosol without the presence of MinD (the concentration of MinE on the membrane, $c_e$, becomes $c_e = 0$ when the total concentration of MinD, $\mathfrak{D}_0$, is $\mathfrak{D}_0 = 0$). This is because, in the absence of MinD, MinE fails to localize to the peripheral portion of the cell. This is in contrast to the observations of MinE binding on the membrane in the absence of MinD in vitro (**Hsieh et al., 2010**; **Park et al., 2017**; **Vecchiarelli et al., 2017**), and with our experiments demonstrating that MinE localization is a key factor in determining Min wave generation.

In both Model I and II, the concentration of MinE on the membrane becomes $c_{e,0}$ in the absence of MinD. The rest of the MinE is in the bulk of the cytosol, and therefore, c/m is given by $(\mathfrak{D}_0 - \alpha c_{e,0})/c_{e,0}$ (see Appendix 2). Therefore, when $c_{e,0}$ is smaller, c/m is larger.

First, we confirmed numerically that the rotating wave occurs in the closed membrane when

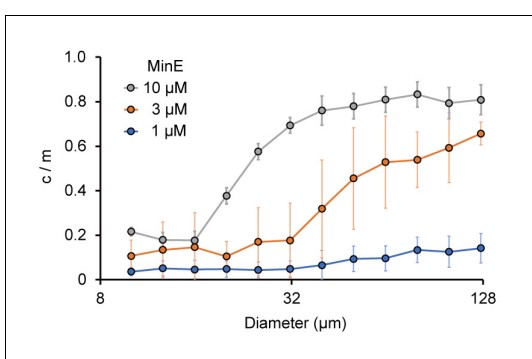

**Figure 4.** Size-dependence of c/m. Spontaneous localization of MinE-mCherry is plotted against size of microdroplets. The average of c/m ratio at each 0.1 logarithmic scale are shown (n = 208 for 1 μM, 386 for 3 μM, and 184 for 10 μM). Error bars indicate standard deviations.
DOI: https://doi.org/10.7554/eLife.44591.018
The following figure supplements are available for figure 4:

**Figure supplement 1.** Localization of MinE near the two-dimensional planar membrane.
DOI: https://doi.org/10.7554/eLife.44591.019
**Figure supplement 2.** Size-dependence of c/m in microdroplets contacting 10 mg/mL BSA.
DOI: https://doi.org/10.7554/eLife.44591.020

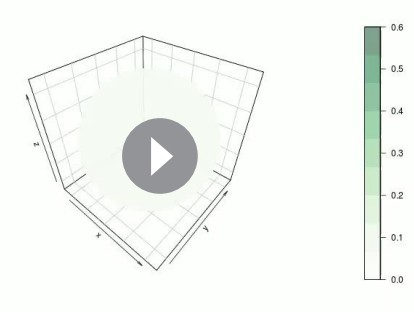

**Video 8.** Time development during the initial stages of MinD single waves in lipid droplets using Model I (simulation).
DOI: https://doi.org/10.7554/eLife.44591.021

MinE localization is weak, $c_{e,0} \sim 0$ (**Video 8**). The wave generation occurred when the total concentrations of MinD and MinE are comparable. We also observed that pole-to-pole oscillation occurs near the boundary between a stationary state and the rotating wave in the phase diagram of the two concentrations. Wave generation on the planar membrane was also observed when MinE localization was weak, consistent with previous studies (**Bonny et al., 2013**; **Halatek and Frey, 2018**).

Next, we studied the effect of spontaneous MinE localization on wave emergence using numerical simulations and linear stability analysis (see 'Materials and methods'). **Figure 5** shows the numerical results describing the amplitude of the wave for the closed membrane in our models (red points for Model II and pink points for Model I). Irrespective to these models, the wave disappeared and the concentrations of MinD and MinE were uniform on the membrane when MinE localization was strong $c_{e,0} \gg 0$. The conditions that are conducive to wave generation may also be evaluated by linear stability analysis of the stationary state. In **Figure 5**, the dark shaded area shows the region in which the stationary state is linearly unstable. In this area, Min waves occurred. Both the numerical results and the linear stability analysis provided evidence that above $c_{e,0} = 0.03$, the Min wave disappears.

Consistent with our experimental results, the numerical simulations indicated that the degree of spontaneous MinE binding shifts the conditions for Min wave emergence (**Figure 5**). We also performed the same analysis for the planar membrane. The linear stability analysis (light shaded area in **Figure 5**) showed that the critical concentration of MinE localization is higher in the planar membrane. Furthermore, the numerical results illustrated an even larger shift of the transition point (as shown by the blue points datapoints in **Figure 5**). These results suggested that the condition is dependent on the size of the membrane; under confinement, the shift is sufficiently strong to eliminate wave generation at stronger MinE localization. On the other hand, wave generation of the planar membrane was less suppressed, and thus, it continued when there was stronger MinE localization on the membrane.

## Theoretical analysis reveals that confinement regulates Min wave emergence

To investigate the effect of confinement, we studied the two models introduced above (Model I and II). We used these two models because they incorporate the two effects (persistent MinE membrane binding and transformation from ADP-MinD to ATP-MinD in cytosol), that were assumed to play essential roles in the wave generation, but had been studied separately. We found that in the closed geometry, these two models reproduce the same results, suggesting that under confinement the difference between the models is not important.

**Figure 6A** shows the phase diagram of the wave generation in the total (membrane plus cytosol) MinD and MinE concentrations according to the linear stability analysis of Model II. The Min wave occurred when both of these concentrations were above the values at the phase boundary for the first mode ($l = 1$). Stability of the stationary state is dependent on the spatially

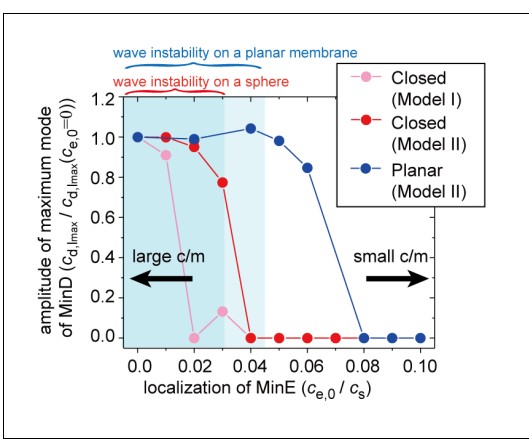

**Figure 5.** Simulation results for wave generation in the presence of membrane-attached MinE. The simulation results for wave generation in the presence of spontaneous MinE interaction at $\mathfrak{D}_0 = 0.5$ and $\varepsilon_0 = 0.8$ (see 'Materials and methods') are shown. Results are shown for the closed spherical membrane and for the planar membrane. Waves are characterized by the amplitude of the first mode ($l = 1$) in spherical harmonics expansion of the closed membrane, and by the maximum amplitude at a finite wave number in the planar membrane. The amplitude is normalized by its value without spontaneous MinE localization. Stability data for the homogeneous state, calculated from the real part of the maximum eigenvalue, show that the homogeneous state is stable in the closed membrane (dark-shaded area) and planar membrane (light shaded area), but linearly unstable otherwise.

DOI: https://doi.org/10.7554/eLife.44591.022

The following figure supplement is available for figure 5:

**Figure supplement 1.** Reaction scheme for computational simulation of Min waves in a cell-sized space.

DOI: https://doi.org/10.7554/eLife.44591.023

inhomogeneous modes. The zero mode ($l = 0$) expressed uniform concentration on the membrane whereas the first mode ($l = 1$) expresses inhomogeneous distribution with one wavelength on the membrane (see *Figure 6A*). The homogeneous oscillation ($l = 0$), in which the concentrations of MinD and MinE are uniform on the membrane but oscillate in time, occurs at another phase boundary shown in *Figure 6A*. The phase boundary of the homogeneous oscillation requires higher concentrations than that of the Min wave, resulting in wave generation rather than uniform oscillation in the closed membrane. This behavior is not obvious in reaction diffusion systems. For any two-variable reaction-diffusion equations, it can be shown that uniform oscillation occurs rather than the wave of the first mode (*Pismen, 2006*). In our models, wave generation did occur by additional degrees of freedom.

To investigate theoretically the mechanism of suppression of the uniform oscillation that results in inhomogeneous wave generation, we considered the generic framework to combine the two models outlined in Appendix 4. Our method enabled us to eliminate the bulk cytosol concentration field. The condition of the wave generation was identified by the real part of the largest eigenvalue $Re\sigma > 0$, where the eigenvalues, $\sigma$, were then obtained by solving the following equation:

$$det\left(\Lambda_l^{(0)} + \mathbf{\Gamma} \cdot \mathbf{M} - \sigma\mathbf{I}\right) = 0 \tag{1}$$

All of the terms in the determinant are $n \times n$ matrices under $n$ concentration fields on the membrane. The first term describes the reaction on the membrane, whereas the second term expressed the effect of bulk cytosol. Here, $\mathbf{I}$ denotes the $n \times n$ identity matrix and $\mathbf{\Gamma}$ shows the coupling of the reactions on the membrane with the bulk cytosol concentrations close to the membrane (see Appendix 4). The effect of confinement in $\mathbf{M}$ appears from its dependence on the size of the system, such as the radius $R$ of a sphere or the height $H$ of the bulk on the planar membrane. *Figure 6B* shows the real part of the largest eigenvalue for the closed membrane as a function of the number of modes. The eigenvalue is positive only at the first mode, suggesting that wave instability occurs instead of uniform oscillation at this mode. This result is independent of choice of Model I or II, and is also independent of the saturation term (see Appendix 3).

From the theoretical analysis, we were able to identify three effects of confinement: (i) the homogeneous stationary solution is dependent on the system size through $\alpha$, (ii) the diffusion on the membrane inhibits higher-mode (smaller length scale) inhomogeneity (see *Equation 32*), and (iii) the effect of the dynamics of the bulk concentrations in *Equation 3* modifies the stability. Among the three contributions, the second one is easily computed once we know the eigenvalues at the zero mode for the matrix:

$$\Lambda_l^{(0)} = \Lambda_{l=0}^{(0)} - \frac{l(l+1)}{R^2}\mathbf{I} \tag{2}$$

In *Figure 6B*, this is demonstrated by the solid lines for each model (see also *Figure 6—figure supplement 1*). It is evident that the stability at the higher modes is dominated by this effect. On the other hand, the eigenvalues at the zero mode deviate from the lines. This result is explained by the effects of (i) and (iii), suggesting that the mechanism of the wave instability is oscillatory instability at the first mode ($l = 1$) with suppression of instability at the zero mode ($l = 0$) due to the effect of confinement.

To see more details about the effect of confinement, we investigated the second term in *Equation 1* (*Figure 7*). For the spherical membrane (*Figure 7A*), the effect of confinement is given by:

$$M_{ij} = \frac{\hat{\xi}_j(\sigma)}{D}\left(p_{ij} + \sigma q_{ij} + \frac{l(l+1)}{R^2}r_{ij}\right)\frac{i_l\left(R/\hat{\xi}_j(\sigma)\right)}{i_l'\left(R/\hat{\xi}_j(\sigma)\right)} \tag{3}$$

where

$$\hat{\xi}_i(\sigma) = \sqrt{\frac{D}{\sigma + \lambda_i}} \tag{4}$$

and $i_l(x)$ is $l$ th-order of the modified spherical Bessel function of the first kind and $i_l'(x) = di_l(x)/dx$.

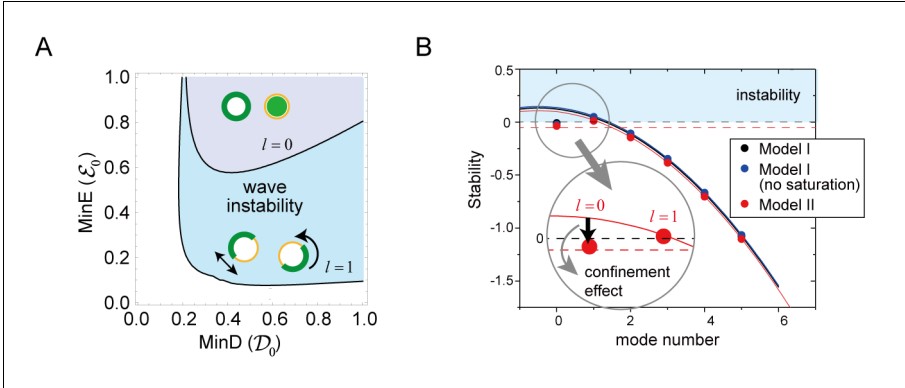

**Figure 6.** Regulation of the generation and stability of Min waves by confinement. (**A**) The phase diagram obtained from linear stability analysis of wave instability (the first mode, $l = 1$) and homogeneous oscillation (the zero mode, $l = 0$) under various total concentrations of MinD and MinE in the closed membrane of size $R = 5$ in Model II. (**B**) Stability of the homogeneous state for each mode obtained by a real part of the eigenvalue in different models near the transition point of wave generation, $\mathfrak{D}_0 = 0.2$ and $\varepsilon_0 = 0.8$. Instability is demonstrated by positive eigenvalues. The black dashed line indicates neutral stability in which the real part of the eigenvalue is zero. The theoretical results, which are an under approximation that neglect the effect of bulk dynamics, are demonstrated by the solid lines (see 'Materials and methods' and *Figure 6—figure supplement 1*). Each color (blue and red) corresponds to a different model. The dashed red line shows the stability of the homogeneous state theoretically obtained by including the effect of confinement.

DOI: https://doi.org/10.7554/eLife.44591.024

The following source data and figure supplement are available for figure 6:

**Source data 1.** Numerical data of *Figure 6B*.

DOI: https://doi.org/10.7554/eLife.44591.026

**Figure supplement 1.** Mode dependence of stability and frequency in the different models obtained from linear stability analysis.

DOI: https://doi.org/10.7554/eLife.44591.025

This effect is significantly different from that on the planar membrane (*Figure 7B*), where the effect of bulk is expressed by:

$$M_{ij} = \frac{\hat{\xi}_j(\sigma)}{D}\left(p_{ij} + \sigma q_{ij} + \frac{l(l+1)}{R^2}r_{ij}\right)\frac{\cosh\left(H/\hat{\xi}_j(\sigma)\right)}{\sinh\left(H/\hat{\xi}_j(\sigma)\right)} \tag{5}$$

where the length scale $\xi$ is expressed by the eigenvalue, $\sigma$, and wave number, $k = |\mathbf{k}|$, as in:

$$\frac{1}{\hat{\xi}_i(\sigma)} = \sqrt{k^2 + \frac{\sigma + \lambda_i}{D}} \tag{6}$$

The magnitude of this effect is represented by $i_l\left(R/\hat{\xi}\right)/i_l'\left(R/\hat{\xi}\right)$ for the closed membrane and $\cosh\left(H/\hat{\xi}\right)/\sin\left(H/\hat{\xi}\right)$ for the planar membrane, both of which are shown in *Figure 7C and D*. As the size $R$ and $H$ decreases, the effect becomes stronger for the zero mode of the spherical membrane and for all of the wave numbers of the planar membrane. This result is in contrast with the higher modes ($l \geq 1$) of the spherical membrane. Thus, for a small system, the effect of the dynamics of bulk remains only for the zero mode of the spherical membrane. The physical picture of this result is that, an inhomogeneous concentration associated with the higher-order modes is suppressed in a small system, while in the planar membrane, inhomogeneity in the plane may exist independently of the direction perpendicular to the membrane.

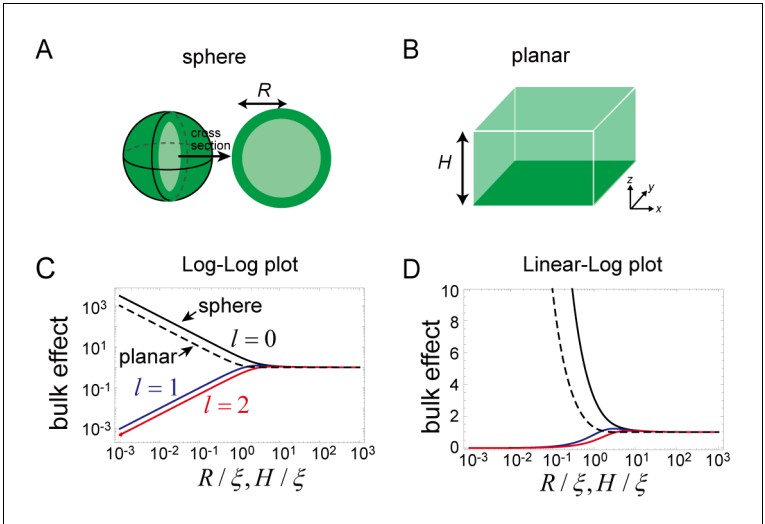

**Figure 7.** Effects of bulk in the models. (**A, B**) Schematic illustration of a closed membrane with the radius $R$ (**A**) and a planar membrane with the height $H$(**B**). Membrane-bound proteins are denoted by dark green color, while bulk proteins are shown in light green. (**C, D**) The log-log (**C**) and log-linear (**D**) plots of $i_l(R/\xi)/i'_l(R/\xi)$ as a function of $R/\xi$. The homogeneous mode ($l = 0$, black) and the two lowest inhomogeneous modes ($l = 1$, blue, and $l = 2$, red) are shown as solid lines. The corresponding term in the planar membrane, $cosh(H/\xi)/sinh(H/\xi)$ as a function of $H/\xi$, is shown as a dashed line.

DOI: https://doi.org/10.7554/eLife.44591.027

## Excitability may occur in the planar membrane

Our numerical simulations suggested that the robustness against MinE localization is stronger than that predicted by the linear stability analysis (*Figure 5*). One possible reason is that the wave generation is dependent on initial perturbation of the concentration fields due to the excitability of the system. The homogeneous stationary state is linearly stable but responds largely against finite perturbation (*Figure 8*). In fact, the instability of the planar membrane starts from a core of wave emergence rather than from uniform oscillation on the planar membrane, as observed in a previous study (*Loose et al., 2008*).

To investigate excitability of the planar membrane under Model II, we first studied the dynamics of the concentrations of membrane-bound proteins without diffusion on the membrane; namely, the bulk concentration was one dimension, and the membrane concentration was zero dimension. At $c_{e,0} = 0.07$ in which the homogeneous stationary state was linearly stable, an initial condition of $c_d$ was shifted from the value at the stationary state $(c_d^*, c_{de}^*, c_e^*) = (0.0105, 0.0750, 0.0708)$, while $c_{de}(t = 0) = c_{de}^*$ and $c_e(t = 0) = c_e^*$. When the deviation, $\delta c_d = c_d(t = 0) - c_d^*$, was small, the system quickly returns to the stationary state. When $\delta c_d > 0.075$, the system initially went away from the stationary state and exhibited a completely different trajectory (*Figure 8A and B*). This behavior suggests that the system is excitable, it is stable against a small perturbation but has a large response to a perturbation above a particular threshold. In contrast with the planar membrane, the closed membrane did not show excitability (*Figure 8B*), and this system quickly relaxed to its stationary state without traveling in a large path. It is known that excitable systems may exhibit dissipative solitary pulses propagating in one direction with fixed speed, and spiral and turbulent waves in two dimensions (*Keener, 1980*; *Bär and Eiswirth, 1993*). In fact, Model II demonstrated a propagating solitary wave when the initial condition was chosen appropriately in one- (*Figure 8C*) and two-dimensional (*Figure 8D*) membranes. A spiral wave was obtained by cutting a solitary band in the two-dimensional membrane (*Figure 8D*), a phenomenon which has also been observed in other excitable systems (*Winfree, 1991*).

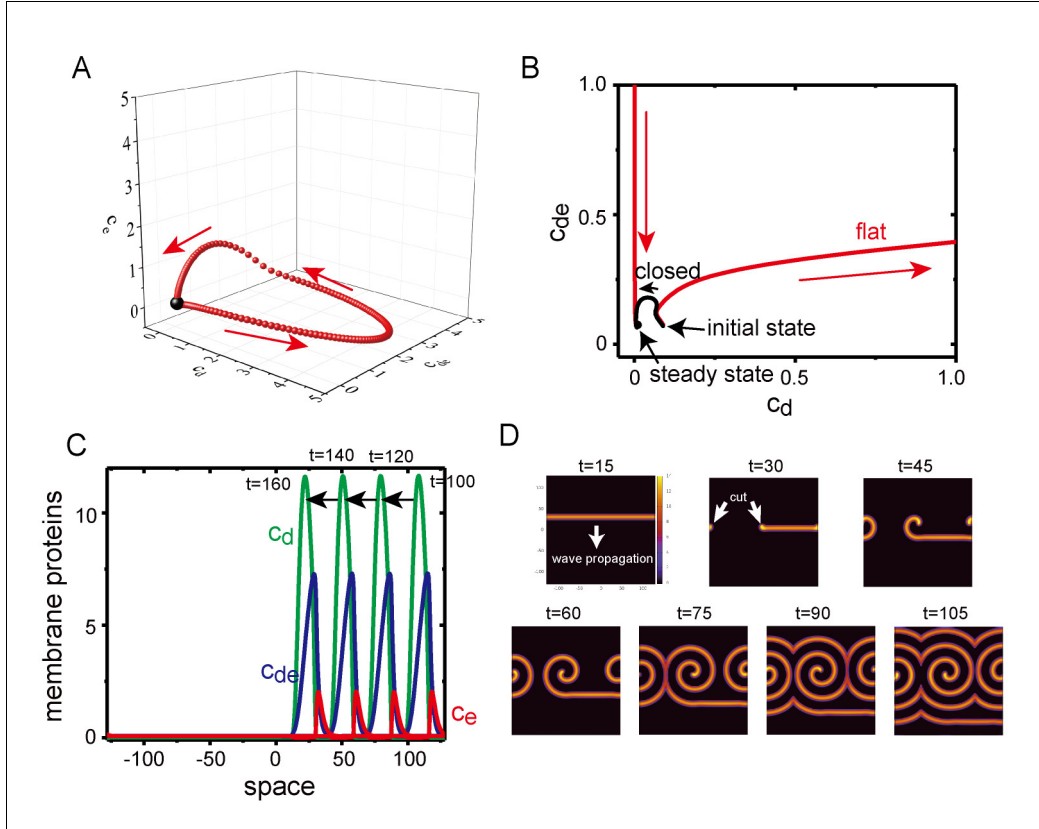

**Figure 8.** Excitability of Min waves on the planar membranes. (**A**) Trajectory of the concentrations of membrane proteins in $(c_d, c_{de}, c_e)$ coordinates starting from the initial condition slightly shifted from the homogeneous stationary state at $\mathfrak{D}_0 = 0.5$, $\varepsilon_0 = 0.8$ and $c_{e,0} = 0.07$. (**B**) Trajectory near the stationary state in the $(c_d, c_{de})$ plane on aplanar membrane ($H = 256$, red) and on a closed membrane ($R = 5$, black). (**C**) A propagating pulse in a one-dimensional membrane surrounded by a two-dimensional bulk. (**D**) A propagating band and a spiral wave in a two-dimensional planar membrane underneath the three-dimensional bulk. Initially, an isolated band is prepared and allowed to propagate, before being cut at $t = 22.5$ to make a spiral wave.

DOI: https://doi.org/10.7554/eLife.44591.028

The following source data is available for figure 8:

**Source data 1.** Numerical data of *Figure 8A*.

DOI: https://doi.org/10.7554/eLife.44591.029

## Early stage of Min wave emergence in microdroplets

Finally, we analyzed the early stage of Min wave emergence in micro-sized space (*Figure 9*, *Video 9*). For small microdroplets that only show a single wave inside, time-lapse imaging of Min proteins showed that pulsing between the cytosolic parts and the membrane surface is the initial stage of Min wave emergence, similar to observations reported previously (*Zieske et al., 2016*). However, our imaging demonstrated that the pulsing pattern transforms to pole-to-pole oscillation, and then, settles into traveling waves. This transition of wave patterns is not specific to wet experiments but can be recapitulated by our computational simulation (*Figure 9*, *Video 8*). The introduction or reduction of stochastic noises to the simulation did not change the results, indicating that this transition proceeds in a deterministic manner; namely, wave instability underlying reaction-diffusion coupling is the only driving force that leads to the emergence of Min waves in the cell-sized space.

## Discussion

Conditions for Min wave appearance have been regarded as being the same for open systems, such as on 2D planar membranes, and for closed geometry, as experienced in a fully confined cell-sized

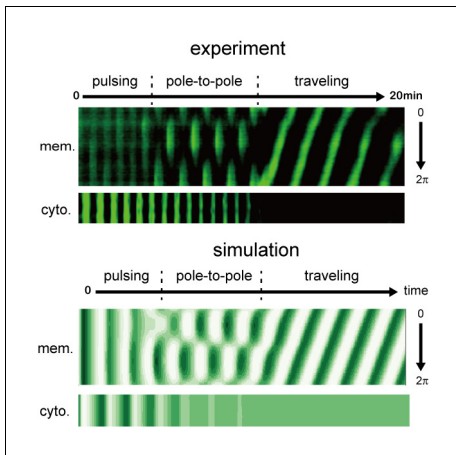

**Figure 9.** Early stages of Min wave emergence inside microdroplets and on 2D membranes. Transition of patterns from pulsing during the initial stage of wave emergence to the formation of a stable traveling wave. Kymographs of sfGFP-MinD in the membranes and inner media of droplets obtained by experiments (top) and numerical simulation without noise (bottom) are shown.

DOI: https://doi.org/10.7554/eLife.44591.030

**Video 9.** The initial stages of the emergence of MinD single waves in lipids droplets.
DOI: https://doi.org/10.7554/eLife.44591.031

space. In this study, we show that the conditions for Min wave appearance are limited in closed cell-sized space as compared with those on a 2D planar membrane. From experiments and simulation, we have shown that the rate of spontaneous localization of MinE is an important factor in determining the generation of Min waves in cell-sized closed spaces..

Owing to the large surface-area-to-volume in the cell-sized space, localization to the membrane (even for weak interactions) becomes more prominent than it is in the planar membrane system. Spontaneous localization of MinE on membranes works in an inhibitory manner with respect to the generation of Min waves, but it is suppressed in the presence of a protein crowder, such as BSA or cell extract, resulting in the generation of Min waves. This effect is observed at a relatively low concentration (cell extract; 1 mg/mL, BSA; 10 mg/mL) relative to the effect of crowding in the cytoplasm of living cells (100–300 mg/mL equivalent) and is not observed with synthetic polymers such as PEG8000 and Ficoll70. Hence, BSA and cell extract are considered to modify the interaction between MinE and membranes, and the mechanism is different from a major effect of crowding, increase of viscosity.

Because a previous report has indicated that BSA at high concentration (>10 mg/mL) attaches to the lipid membrane (*Ruggeri et al., 2013*), and as several proteins in cell extract are assumed to interact with such membranes, a plausible explanation of the effect of the protein crowders is competitive inhibition. To match this assumption, tuning lipids conditions to reduce spontaneous MinE attachment on membranes (*Figure 3F*) seems to be important for the generation of Min waves without aid from auxiliary molecules, as reported previously (*Zieske et al., 2016*; *Litschel et al., 2018*). Although estimation of the exact strength of interaction between MinE and membranes in the cell-sized space is important for understanding the details of the spontaneous membrane binding of MinE, we failed to achieve this because of the technical difficulties. However, the level is assumed to be weak from a previous study using 2D planar membranes (1/100 of the strength of MinD binding) (*Vecchiarelli et al., 2017*).

Recent studies have suggested that the conformation of MinE is in equilibrium between a free state of membrane targeting sequences (MTS) at the N-terminal (open conformation) and a packed structure (closed conformation). Open conformation preferably binds membranes, and several MinE mutants shift this equilibrium to the open state (*Park et al., 2017*; *Denk et al., 2018*). Thus, another possible mechanism to suppress the membrane attachment of MinE on membranes is regulation of the open-closed equilibrium state by excluded volume or other effects that we did not consider. This point will be clarified by analyzing the open-closed equilibrium state of MinE in a similar manner to that used in a previous study (*Park et al., 2017*) in the presence of protein crowders.

Our computational simulations showed that Min wave emergence depends on membrane MinE accumulation and membrane size (*Figure 5*). Cell-sized space stabilized the homogenous state (*Figure 6*), and therefore, Min waves, which are inhomogeneous oscillations in space, emerge instead of homogeneous oscillations. By contrast, for the planar membrane, the wave robustly appears despite the increase of the spontaneous MinE binding. Our simulation results suggest that this robustness is explained by the two mechanisms: coupling between the membrane and bulk dynamics, and the

excitability of the system. If the system is excitable, the homogeneous stationary state is linearly stable, but responds strongly to finite perturbation (*Figure 8*). We stress that excitability occurs only in the planar membrane because in a closed small membrane, any large change in the local membrane concentrations during an excitable wave inevitably changes the bulk concentrations. Although our simulation suggests that the effects of excitability are stronger than space size effects, it remains to be determined whether or not the excitability shown by this Min wave is model independent. This question would be clarified by investigation of the wave generation under controlled initial conditions in further experiments.

Recent in vitro reconstitution studies have demonstrated that biosystems in cell-sized spaces show characteristic features of those biosystems that are not found in test tubes. For example, a cell-sized space enhances the formation of the actomyosin ring (*Miyazaki et al., 2015*), affects aqueous phase separation (*Yanagisawa et al., 2014a*; *Yanagisawa et al., 2014b*), and confers scaling properties on spindle shapes (*Good et al., 2013*). Although these studies have determined that space size is a cue to change the behaviors of biosystems, the biochemical parameters and mechanisms underlying these behaviors have been assumed to be equal irrespective of space sizes. Our present study provides evidence that cell-sized space shifts the equilibrium of the membrane binding of proteins, and changes the conditions that allow the generation and stability of iRD waves from those seen in 2D planer membranes. As theoretical analysis of Min wave behaviors have indicated, the control of wave generation and stability by cell-sized confinement are expected to be universal features among iRD waves. Furthermore, equilibrium shifts in protein localization resulting from surface-to-volume effects should be universal among biosystems because maximum attachment levels and total amounts of the factor explained the shift. These points will be elucidated by in vitro reconstitution of other iRD systems (*Adachi et al., 2006*; *Goryachev and Pokhilko, 2008*; *Arai et al., 2010*; *Huang et al., 2013*).

## Materials and methods

### Expression and purification of MinD and its mutant

In this study, all *Escherichia coli* cells were cultivated in LB medium. To construct pET15b-MinD, the MinD gene was cloned from the *E. coli* MG1655 genome by PCR into pET15b (Merck Millipore, Billerica, MA, USA) by Gibson assembly (New England Biolabs, Ipswich, MA, USA). To construct pET15-sfGFP-minD, the sfGFP gene was amplified from pET29-sfGFP (*Fujiwara and Doi, 2016*) by PCR and cloned into MinD gene by Gibson assembly. It is transcribed to insert amino acids into the N-terminal of MinD. To construct pET15-sfGFP-MinD$^{D40A}$Δ10, the D40A mutation, which results in the adeletion of the C-terminal 10 amino acids of the MinD protein, were introduced into pET15-sfGFP-minD by using the PrimeSTAR Max mutagenesis protocol (TaKaRa, Shiga, Japan). Similarly, the K11A mutation was introduced into the pET15-sfGFP-minD construct in order to produce pET15-sfGFP-MinD$^{K11A}$. *E. coli* BL21-CodonPlus(DE3)-RIPL (Agilent Technologies, Santa Clara, CA, USA) cells were transformed with the resultant plasmids.

Protein expression was induced by 1 mM isopropyl β-D-1-thiogalactopyranoside (IPTG) at $OD_{600}$ = 0.1–0.2 and further cultivation at 37°C for 3 to 4 hr. The cells were collected by centrifugation and suspended in LS buffer [50 mM $NaH_2PO_4$ (pH 7.6), 300 mM NaCl, 10 mM imidazole, 10 mM dithiothreitol (DTT), 0.1 mM phenylmethylsulfonyl fluoride (PMSF)] with 0.2 mM ADP-Mg. The collected cells were disrupted by sonication using a Sonifier250 (Branson, Danbury, CT, USA), and the supernatant of the crude extract was fractionated by centrifugation at 20,000 g at 4°C for 30 min. To purify His-tagged proteins, the crude extracts mixed with cOmplete His-Tag purification resin (Roche, Basel, Switzerland) were loaded onto a polyprep chromatography column (Bio-Rad, Hercules, CA, USA), and washed with 25 mL WS buffer [50 mM $NaH_2PO_4$ (pH 7.6), 300 mM NaCl, 20 mM imidazole, 10% glycerol, 0.1 mM EDTA, and 0.1 mM PMSF]. His-tagged proteins were eluted with EL buffer [50 mM $NaH_2PO_4$ (pH 7.6), 300 mM NaCl, 250 mM imidazole, 10% glycerol, 0.1 mM EDTA, and 0.1 mM PMSF]. EL buffer was exchanged with storage buffer [50 mM HEPES-KOH (pH 7.6), 150 mM GluK, 10% glycerol, 0.1 mM EDTA] with 0.2 mM ADP-Mg by ultrafiltration using AmiconUltra-15 10 k and AmiconUltra-0.5 30 k filters (Merck Millipore).

For pull-down assay, His-sfGFP-MinD$^{D40A}$Δ10 was treated with thrombin (Wako, Osaka Japan) in the storage buffer at 4°C overnight. Then, the cleaved His-Tag (2 kDa) was removed from the sfGFP-

MinD$^{D40A}$Δ10 (55 kDa) solution by ultrafiltration using AmiconUltra-0.5 50 k filters (Merck Millipore). Proteins in the storage buffer were stored at −80˚C. Protein purity and concentrations were estimated by Comassie Brilliant Blue (CBB) staining after separating by sodium dodecyl sulphate polyacrylamide gel electrophoresis (SDS-PAGE) and bicinchoninic acid (BCA) assay.

## Expression and purification of MinE

To construct pET29-minE-His and pET29-minE-mCherry-His, MinE and mCherry genes were amplified from the *E. coli* K12 MG1655 genome and the pET21b-RL027A plasmid (Addgene, Cambridge, MA, USA), respectively, and were cloned into pET29a (Merck Millipore) by Gibson assembly. The 6xHis-tag at the C-terminal of MinE or mCherry was attached by PCR. *E. coli* BL21-CodonPlus (DE3) RIPL cells were transformed with the resultant plasmids.

Proteins were expressed after induction by 1 mM IPTG at $OD_{600}$ = 0.1–0.2 and further cultivation at 37˚C for 3 to 4 hr (pET29-minE-His) or at 16˚C for 12 hr (pET29-minE-mCherry-His). Cells were collected by centrifugation, resuspended in LS buffer, and purified using the protocol described for MinD. The elution fraction of MinE-mCherry-His diluted 5- to 10-fold with HG buffer (50 mM HEPES-KOH, pH 7.6, 10% glycerol, and 0.1 mM EDTA] was further purified by using HiTrap Q HP column (GE Healthcare, Chicago, IL, USA) and AKTA start (GE Healthcare). Briefly, the diluted fraction was loaded onto the column equilibrated with buffer A [50 mM HEPES-KOH (pH 7.6), 50 mM NaCl, 10% glycerol, and 0.1 mM EDTA], and washed using the same buffer. Proteins were eluted using the IEX protocol of AKTA start using buffer A and buffer B [50 mM HEPES-KOH (pH 7.6), 1 M NaCl, 10% glycerol, and 0.1 mM EDTA]. Peak fractions monitored by SDS-PAGE were collected and exchanged with the storage buffer using AmiconUltra-15 10 k and AmiconUltra-0.5 10 k filters (Merck Millipore). Samples were stored at −80˚C, and protein purity and concentrations were estimated by CBB staining after separation by SDS-PAGE and BCA assay. In the case of MinE-mCherry-His, concentrations were estimated by quantitative CBB staining using Fiji software (National Institutes of Health, Bethesda, MD, USA) to avoid signal contamination from mCherry absorbance.

## Expression and purification of sfGFP-MinC

The MinC gene and sfGFP gene were amplified and cloned into the pET15b vector using the the same procedure that was used for MinD. *E. coli* BL21-CodonPlus(DE3)-RIPL cells were transformed with the resultant plasmid. IPTG was added at $OD_{600}$ = 0.1–0.2 to 1 mM, and cells were further cultivated at 16˚C overnight. The protocols for the purification, storage, and quantification of His-sfGFP-MinC were the same as those used for MinD except that no ADP-Mg was added.

## Preparation of *E. coli* cell extract

*E. coli* BL21-CodonPlus(DE3)-RIPL cells were cultured in LB medium at 37˚C. Cells at $OD_{600}$ = 0.7 were collected by centrifugation and suspended in LSE buffer [25 mM Tris-HCl (pH 7.6), 250 mM NaCl, and 10 mM GluMg]. Then, cells were disrupted by sonication using the Sonifier250, and the supernatant of the crude extract after centrifugation at 30,000 g for 30 min at 4˚C was collected as cell extract. To remove genome DNA and RNA, cell extract was incubated at 37˚C for 30 min. The supernatant after centrifugation at 30,000 g for 30 min at 4˚C was exchanged with the RE buffer [25 mM Tris-HCl (pH 7.6), 150 mM GluK and 5 mM GluMg] using AmiconUltra-15 3 k and AmiconUltra-0.5 3 k filters (Merck Millipore). The sample was stored at −80˚C, and protein concentration was estimated by BCA assay. Concentrations of total RNA, includingribosomal RNA, tRNA, and mRNA, were estimated by 260 nm absorbance. Macromolecule concentrations were determined by the summation of protein and RNA concentration (*Fujiwara and Nomura, 2013*).

## Preparation of supported lipid bilayers (SLBs) on a mica layer

The general protocol from a previous report (*Vecchiarelli et al., 2014*) was followed. *E. coli* polar lipid extract (Avanti, Alabaster, AL, USA) in chloroform at 25 mg/mL was dried by argon gas flow. The lipid film was further dried in a desiccator for at least 30 min at room temperature, followed by resuspension in TKG150 buffer [25 mM Tris-HCl (pH 7.6) and 150 mM GluK] to a lipid concentration of 5 mg/mL and then gentle hydration at 23˚C for at least 1 hr. The lipid solution was then vortexed for 1 min and sonicated using the Sonifier250 for 10–15 min (Duty10%, Output1) to obtain small unilamellar vesicles (SUVs). The SUV solution was diluted to 2 mg/mL with TKG150 buffer, and $CaCl_2$

was added to a final concentration of 0.1 mM. This solution was applied to a thin mica layer mounted on the bottom of a glass base dish (Iwaki, Tokyo, Japan). After a 1 hr incubation at 37°C, excess SUVs were washed with RE buffer.

## Self-organization assay for Min proteins on SLBs

For the self-organization assay, a reaction mixture containing 2.5 mM ATP, 1 μM His-sfGFP-MinD, and 1 μM MinE-mCherry-His in RE buffer was added to the SLBs, followed by incubation at room temperature for 10 min prior to microscopic observation. Self-organization of Min proteins was observed using a fluorescent microscope (Axiovert 200M; Carl Zeiss, Jena, Germany) with a CMOS camera using an ORCA-Flash4.0 V2 (Hamamatsu Photonics, Shizuoka, Japan) or a confocal laser-scanning microscope FV1000 (Olympus, Tokyo, Japan).

## Self-organization assay inside lipid droplets

The general protocol for microdroplets preparation was followed according to a previous report (*Fujiwara and Yanagisawa, 2014*). *E. coli* polar lipid extract (Avanti) in chloroform at 25 mg/mL was dried by argon gas flow and dissolved in mineral oil (Nacalai Tesque, Kyoto, Japan) to 1 mg/mL in glass tubes. The lipid mixture was then sonicated for 90 min at 60°C using Bransonic (Branson). For preparation of the modified lipid mixture, 15% of 10 mg/mL *E. coil* cardiolipin (CA) (Avanti) and 85% of 10 mg/mL 1,2-dioleoyl-sn-glycero-3-phosphocholine (DOPC) (Avanti) dissolved in chloroform were mixed and microdroplets were prepared as the same way as for *E. coli* polar lipid extract. For the self-organization assay, the reaction mixture consisted of 1 μM His-sfGFP-MinD, 1 μM MinE-mCherry-His, 2.5 mM ATP, and macromolecules [BSA of Cohn Fraction V (A6003, Sigma-Aldrich, St. Louis, MO, USA), *E. coli* cell extract, Ficoll70 (Santa Cruz Biotechnology, Dallas, TX, USA), or PEG8000 (Promega, Madison, WI, USA) in RE buffer]. In the case of the MinC assay, 1 μM His-MinD, 1 μM MinE-His, and 0.3 μM His-sfGFP-MinC were used. Concentrations of BSA and *E. coli* cell extract were varied to evaluate the concentration dependence of Min waves. To avoid depletion of ATP due to the presence of endogenous enzymes in the *E. coli* cell extract, 80 mM creatine phosphate and 0.4 mg/mL creatine kinase were added for the assay using cell extract. The reaction mixture (2 μL) was added to the lipid mixture (100 μL), and lipids microdroplets were obtained by emulsification with tapping. A portion of the mixture (15 μL) was gently placed into two glass coverslip slits with a double-sided tape as spacers. Self-organization of Min proteins inside the droplets was observed using the same equipment described for SLBs.

## Diffusion analysis

A confocal laser-scanning microscope was used (FV1200; Olympus) for analysis of the diffusion of sfGFP in cytosolic parts and of His-sfGFP-MinD on membranes in BSA solution entrapped inside microdroplets covered with *E. coli* polar lipids. The diffusion coefficient of sfGFP in 0 mg/mL, 50 mg/mL, 100 mg/mL, 200 mg/mL, and 300 mg/mL of BSA in RE buffer was measured by the standard protocol for Fluorescence Correlation Spectroscopy of FV1200. The diffusion coefficients of His-sfGFP-MinD (1 μM) and MinE-mCherry-His (1 μM) on membranes with or without 100 mg/mL of BSA in RE buffer were measured by fluorescence recovery after photo-bleaching (FRAP) using tornado bleaching of a circular area with ~1 μm diameter. The recovery intensity as a function of time was converted to the diffusion coefficients using the FRAP protocol of FV1200.

## Pull-down assay

The mixture of 9 μM MinE-mCherry-His, 3 μM His-sfGFP-MinD, or 6 μM MinD mutant treated by thrombin (MinD$^{D40A}$Δ10), and 3 μM BSA were applied to cOmplete His-Tag purification resin and incubated in RE buffer for 30 min at room temperature. Each mixture with resin was loaded into Micro Bio-Spin chromatography columns (Bio-Rad). Then, flow-thorough fraction was separated and collected by a tabletop centrifuge. After washing the resin by 500 μL RE buffer with 20 mM imidazole for 3–5 times, elution fraction was obtained by 50 μL RE buffer with 250 mM imidazole. Proteins in each fraction were separated by SDS-PAGE and visualized by CBB staining.

## Evaluation of c/m ratio

Preparation of lipid droplets and glass coverslips for observation were performed using the same procedure as was used for the self-organization assay inside lipid droplets. To analyze the localization (c/m) of MinE, various concentrations of MinE-mCherry-His were mixed with BSA. For the analysis using *E. coli* cell extract, Ficoll70 (Santa Cruz Biotechnology, Dallas, TX, USA) or PEG8000 (Promega, Madison, WI, USA), 1 μM MinE-mCherry-His was used. Then, the mixture was entrapped inside microdroplets of *E. coli* polar lipids. To analyze the localization of other proteins, 1 μM sfGFP-MinD with or without 50 mg/mL BSA, 1 μM sfGFP, or 5 μM BSA-FITC (Thermo Fisher Scientific, Waltham, MA, USA) with 5 μM BSA were used. The membrane localization of each protein was observed using a confocal laser-scanning microscope FV1000 (Olympus). All analyses of obtained images were carried out using Fiji software. A center line of a droplet was manually drawn, and then the intensity of each pixel along that line was obtained. The membrane intensities (m) were determined as the higher intensity of two membrane edge peaks. The cytosol intensities (c) were determined as the average intensity of 10 pixels around the pixel at the center position between two edge peaks. To evaluate the effects of macromolecules (Ficoll70, and PEG8000) and the modified lipid condition (85% of DOPC and 15% cardiolipin) in *Figure 3F, c/m* ratio was determined as the average from 10 individual droplets of 10–30 μm in diameter.

## Numerical simulations

The partial differential equations in Model I (*Equations 12 and 16*) and Model II (*Equations 19 and 24*) were solved either using the commercial software of the Finite Element method, COMSOL, or using custom codes of the pseudo-spectral method in the spherical coordinates $(r, \theta, \varphi)$ for the closed membrane and Cartesian coordinates $(x, y, z)$ for the planar membrane. Both methods reproduce waves in the planar membrane and in the closed membrane. In the pseudo-spectral method for the closed membrane of a spherical shape, all the concentration fields in bulks such as $c_D$ and on a membrane such as $c_d$ are expanded in terms of spherical harmonics, $Y_l^m(\theta, \varphi)$:

$$c_D(r, \theta, \varphi, t) = \sum_{l=0}^{l_{max}} \sum_{m=-l}^{l} c_{D,lm}(r, t) Y_l^m(\theta, \varphi) \tag{7}$$

$$c_d(\theta, \varphi, t) = \sum_{l=0}^{l_{max}} \sum_{m=-l}^{l} c_{d,lm}(t) Y_l^m(\theta, \varphi). \tag{8}$$

The model was then translated into a set of ordinary differential equations to obtain membrane concentrations $(c_d, c_{de}, c_e)$ and one-dimensional partial differential equations (time, $t$, and the radial direction, $r$) for the bulk concentrations $(c_D, c_E)$. In total, we solved $(l_{max} + 1)^2$ equations for each variable where the truncation of the mode was chosen as $l_{max} = 16$. The results were independent of the increase in the value.

Inhomogeneity of the concentration field on the membrane was expressed by the amplitude of each mode denoted by $l$. The amplitude is expressed by a rotationally invariant form using the expansion coefficients in *Equation 7*, with all $m \in [-l, l]$. For example, the uniform distribution of MinD on the membrane was expressed by the $l = 0$ mode and its norm $\sqrt{c_{d,00}^2}$, while the first mode ($l = 1$) corresponds to the inhomogeneous concentration field of a single wave, which is characterized by the norm $\sqrt{c_{d,1,0}^2 - c_{d,1,-1}c_{d,1,1}}$.

The parameters were set as $\omega_D = 0.1$, $\omega_{dD} = 5.0$, $\omega_E = 0.1$, $D = 100$, and $\omega_{ed} = 100$ in the non-dimensional unit (see Appendix 2). If we choose $\omega_e = 0.2[1/\text{sec}]$, $D_d = 0.2[\mu m^2/\text{sec}]$, and the units of concentrations in the cytosol and on the membrane to be $10^3[1/\mu m^3]$ and $10^3[1/\mu m^2]$, respectively, then, our choice of the parameters implies $\omega_D = 0.02$ [μm/sec], $\omega_{dD} = 10^{-3}$ [μm³/sec], $\omega_E = 2 \times 10^{-5}$ [μm³/sec], $D = 20$ [μm²/sec], and $\omega_{ed} = 2 \times 10^{-2}$ [μm²/sec]. MinE localization at the membrane was modeled by the term $c_{e,0}$. When BSA was added, we set smaller $c_{e,0}$, whereas without BSA, we set $c_{e,0} > 0$. Note that when $c_{e,0} = 0$, all MinE molecules are in bulk $c_e = 0$ without MinD, while the membrane is filled by MinD, that is $c_d = 1$, without MinE.

For the planar membrane in the $\mathbf{x} = (x, y)$ plane, the concentration fields are expanded with the wave vector, $\mathbf{k}$, such as

$$c_D(h, \mathbf{x}, t) = \int_{\mathbf{k}} c_{D,\mathbf{k}}(z, t) e^{i\mathbf{k} \cdot \mathbf{x}} \text{ and} \tag{9}$$

$$c_d(\mathbf{x}, t) = \int_{\mathbf{k}} c_{d,\mathbf{k}}(t) e^{i\mathbf{k} \cdot \mathbf{x}}, \tag{10}$$

with the wave-number-dependent expansion coefficients $c_{d,k}(t)$ on the membrane and $c_{D,k}(z, t)$ in the bulk cytosol. Here, the amplitude of the wave vector is denoted by the wave number, $k = |\mathbf{k}|$. We may use the pseudo-spectral method, and solve the fields in the direction of the height, $z$, in real space, and the fields in the direction of the plane, $(x, y)$, in Fourier space. The amplitude of a wave of MinD on the membrane is given by the absolute value of the complex number of the expansion coefficient $|c_{d,k}|$.

## Stability analysis of the theoretical models

We performed linear-stability analysis on the models. First, we calculated the stationary uniform solutions of the equations by setting time and spatial derivatives along the direction on the membrane to zero, and denoted these solutions by superscript '*'. *Equations 12 and 16* and the boundary conditions [*Equations 17 and 18*] were then linearized around the stationary uniform solution, such as $c = c^* + \delta c$. The eigenvalues, $\sigma$, are obtained by plugging $\delta c(t) = \delta c e^{\sigma t}$ into the linearized equations (*Pismen, 2006*; *Gou et al., 2015*). The partial differential equations for the bulk dynamics were solved and the boundary conditions were translated into linear relationship between membrane and bulk concentrations. The set of the linearized equations for the concentration fields, for example $\boldsymbol{\Psi} = (c_d, c_{de}, c_e, c_D, c_E)$ in Model I, is expressed by a matrix form as

$$\sigma \mathbf{I}_s \cdot \delta \boldsymbol{\Psi} = \Lambda_l \cdot \delta \boldsymbol{\Psi}, \tag{11}$$

where the $5 \times 5$ matrix $\Lambda_l$ has five eigenvalues depending on the mode $l$ (but not on $m$) of spherical harmonics for the closed membrane. Here, $\mathbf{I}_s$ shows the dynamics on the membrane and is a diagonal matrix whose diagonal elements are one only for the membrane concentrations and 0 otherwise, for example in Model I $(1, 1, 1, 0, 0)$. The concentration in bulk in $\boldsymbol{\Psi}$ is interpreted as the concentration near the membrane, such that $c_D(R, \theta, \varphi)$ for the closed membrane and $c_D(x, y, 0)$ for the planar membrane. For the planar membrane, the matrix is dependent on the wave number $k$ and is denoted by $\Lambda_k$. When the real part of the eigenvalue is positive, that is $Re\Lambda_l > 0$ for $l \neq 0$, the uniform state is unstable, and an inhomogeneous pattern appears. In addition, when the imaginary part is non-zero, the frequency becomes finite and either standing or rotating waves appear. In Model II, the same analysis was performed for the concentration fields denoted by $\boldsymbol{\Psi} = (c_d, c_{de}, c_e, c_{DT} + c_{DD}, c_{DD}, c_E)$ and the $6 \times 6$ matrix $\Lambda_l$ in (*Equation 11*).

## Acknowledgements

We thank Ms. A Yoshida (Keio University) for supporting protein purification, and Prof. K Yoshikawa (Doshisha University), Prof. H Kitahata (Chiba University), Prof. T Sakurai (Chiba University), Prof. Karsten Kruse (University of Geneva), and Prof. Toshiyuki Ogawa (Meiji University) for helpful discussion. We are grateful for financial support from JSPS KAKENHI Grant Number JP16H00809, JP26650044, JP15KT0081, JP15H00826, JP18H04565 (to KF), JP26800219, JP16H00793, and JP17K05605 (to NY). We are also grateful for a Ph.D. Program Research Grant at Keio university awarded to SK.

## Additional information

### Funding

| Funder | Grant reference number | Author |
| --- | --- | --- |
| Japan Society for the Promotion of Science | JP16H00809 | Kei Fujiwara |
| Japan Society for the Promotion of Science | JP26800219 | Natsuhiko Yoshinaga |
| Japan Society for the Promotion of Science | JP26650044 | Kei Fujiwara |
| Japan Society for the Promotion of Science | JP15KT0081 | Kei Fujiwara |
| Japan Society for the Promotion of Science | JP15H00826 | Kei Fujiwara |
| Japan Society for the Promotion of Science | JP18H04565 | Kei Fujiwara |
| Japan Society for the Promotion of Science | JP16H00793 | Natsuhiko Yoshinaga |
| Japan Society for the Promotion of Science | JP17K05605 | Natsuhiko Yoshinaga |

The funders had no role in study design, data collection and interpretation, or the decision to submit the work for publication.

### Author contributions

Shunshi Kohyama, Resources, Investigation, Visualization, Methodology, Writing—original draft, Performed all wet experiments, Conceived wet experiments with KF; Natsuhiko Yoshinaga, Resources, Formal analysis, Funding acquisition, Investigation, Visualization, Methodology, Writing—original draft, Writing—review and editing, Conceived, designed and performed the simulation and theoretical analysis; Miho Yanagisawa, Resources, Supervision, Writing—review and editing; Kei Fujiwara, Conceptualization, Resources, Funding acquisition, Visualization, Methodology, Writing—original draft, Project administration, Writing—review and editing, Conducted and designed the research, Conceived wet experiments with SK; Nobuhide Doi, Supervision

### Author ORCIDs

Shunshi Kohyama (ID) https://orcid.org/0000-0002-7588-2488
Natsuhiko Yoshinaga (ID) https://orcid.org/0000-0001-9434-7010
Miho Yanagisawa (ID) https://orcid.org/0000-0001-7872-8286
Kei Fujiwara (ID) https://orcid.org/0000-0001-7308-774X

### Decision letter and Author response

Decision letter https://doi.org/10.7554/eLife.44591.038
Author response https://doi.org/10.7554/eLife.44591.039

## Additional files

### Supplementary files

• Transparent reporting form
DOI: https://doi.org/10.7554/eLife.44591.032

### Data availability

All data generated or analyzed during this study are included in the manuscript and supporting files. Source data files have been provided for Figures 1E, 6B, and 8A.

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

## Appendix 1

DOI: https://doi.org/10.7554/eLife.44591.033

# Theoretical analysis of the Min systems of closed and planar membranes

Before explaining the theoretical models in this study in Appendix 2 and 3, we summarize current theoretical understanding of Min wave generation. Spontaneous generation of waves, observed in experiments, have been studied as wave instability in reaction-diffusion equations. This approach was first proposed by Alan Turing as an extension of static Turing instability (*Turing, 1952*). In contrast to the static instability realized by a minimum of two components, wave instability requires three components. Although the instability can be evaluated by the linear stability analysis, physical intuition of the wave instability is not as obvious as static Turing instability. Therefore, several mechanisms have been proposed: first, homogeneous oscillation in a two-component system is suppressed by a third component (*Gourley and Britton, 1996*); second, a static inhomogeneous pattern generated by two-component Turing instability becomes oscillatory due to a third component (*Okuzono and Ohta, 2003*); and, third, oscillatory instability occurs in eigenmodes that are associated with conserved quantities (*Kessler and Levine, 2016*).

Despite these hypotheses, a clear mechanism of wave generation within the Min system still remains under debate due to the complexity of reaction couplings, and here is also a lack of understanding about how the mixture of different spatial dimensions (membrane and cytosol) plays a role. The Min systems have two representative geometries: one is the closed membrane (*Figure 7A*), which we focused in this study, and the second is the open planar membrane (*Figure 7B*). In both systems, MinD and MinE proteins are distributed on the two-dimensional membrane and in the three-dimensional bulk cytosol.

Recent theoretical studies have focused on realistic computational simulations to reveal the mechanism of Min waves. These studies are based on partial differential equations of reaction-diffusion models (*Howard et al., 2001*; *Meinhardt and de Boer, 2001*; *Huang et al., 2003*; *Huang and Wingreen, 2004*; *Halatek and Frey, 2012*; *Bonny et al., 2013*; *Kessler and Levine, 2016*; *Denk et al., 2018*; *Halatek and Frey, 2018*) or particle-based stochastic models (*Fange and Elf, 2006*; *Kerr et al., 2006*; *Arjunan and Tomita, 2010*; *Hoffmann and Schwarz, 2014*). The stochastic model is based on the model proposed by *Huang et al. (2003)*. At an early stage of the modeling, the geometry of the Min system was neglected, such that all of the concentration fields, both on the membrane and in the bulk cytosol, were defined in the same dimensions (*Meinhardt and de Boer, 2001*; *Howard et al., 2001*). Recently, the coupling between the dynamics of membrane and bulk have been investigated. Among these models, only two approaches have successfully reported the reproduction of the Min wave generation on both the planar and closed membranes, including the effect of Min proteins in the cytosol. One approach is to include the transformation from ADP-MinD to ATP-MinD (*Huang et al., 2003*), and the second is to include the formation of a MinDE complex from membrane-bound MinD and MinE, resulting in persistent MinE membrane binding (*Bonny et al., 2013*). In both models, the Min wave occurs in the planar membrane (*Bonny et al., 2013*; *Halatek and Frey, 2018*). The wave on the closed membrane requires a specific initial condition, stochasticity (*Fange and Elf, 2006*) or ellipsoidal shape (*Halatek and Frey, 2012*), when the model proposed by *Huang et al. (2003)* is used, whereas the wave occurs without these effects in *Bonny et al. (2013)*. The two approaches show different dependence of the stability of the waves on total MinD and MinE concentrations, and other parameters.

# Appendix 2

DOI: https://doi.org/10.7554/eLife.44591.033

## Theoretical model I

In Model I, MinD and MinE concentrations inside a spherical membrane with radius $R$, or in the rectangular bulk with its height $H$, were denoted by $c_D$ and $c_E$, respectively (see **Figure 7A and B**). Concentrations of MinD, MinE, and their complex (MinDE) bound to a membrane were denoted by $c_d$, $c_e$, and $c_{de}$, respectively (see **Figure 5—figure supplement 1**). The total MinD and MinE concentrations are denoted by $\mathfrak{D}_0 = c_D + \alpha(c_d + c_{de})$ and $\varepsilon_0 = c_E + \alpha(c_{de} + c_e)$, respectively. We denote the characteristic concentrations on the membrane and in the cytosol as $c_s$ and $c_b$, respectively, and we express all of the concentration fields in the unit of these characteristic concentrations. Here, $\alpha$ demonstrates an effect of confinement. Its concrete form is dependent on the geometry of the system, but, in the current model for a spherical closed membrane, $\alpha = 3c_s/(c_b R)$. For the planar membrane, it is associated with the height $H$ of the system as $\alpha = c_s/(c_b H)$.

Chemical reactions are schematically shown in **Figure 5—figure supplement 1A**. Each reaction shows a rate, $\omega$, specified by its subscript. The diffusion constants of proteins bound to the membrane were denoted by $D_d$, $D_e$, and $D_{de}$, whereas the bulk diffusion of unbound proteins was denoted by the diffusion constants $D_D$ and $D_E$. We assumed the same diffusion constants for MinD and MinE in bulk represented by $D$. We also assumed the same diffusion constants for $D_d$, $D_e$, and $D_{de}$ on the membrane. The latter diffusion constant was chosen to be unity without loss of generality. In comparison to the original work in **Bonny et al. (2013)**, the unbinding process was approximated as $\omega_{de,m} = \omega_{de} \approx \omega_e$, and $\omega_{de,c} = 0$. We defined the unit time scale as $\tau_0 = 1/\omega_e$ and the unit length scale as $l_0 = \sqrt{D_d/\omega_e}$. The concentration fields on the membrane were normalized by the characteristic concentration on the membrane, $c_s$, which is chosen to be the maximum concentration on the membrane, $c_{max}$, in the presence of the saturation effect (Model I). The model is given by the following:

Model I

$$\partial_t c_D = D\Delta c_D \tag{12}$$

$$\partial_t c_E = D\Delta c_E \tag{13}$$

$$\partial_t c_d = \Delta_s c_d + c_D(\omega_D + \omega_{dD} c_d)(1 - c_d - c_{de}) - \omega_E c_E c_d - \omega_{ed} c_e c_d \tag{14}$$

$$\partial_t c_{de} = \Delta_s c_{de} + \omega_E c_E c_d + \omega_{ed} c_e c_d - c_{de} \tag{15}$$

$$\partial_t c_e = \Delta_s c_e + c_{de} - \omega_{ed} c_e c_d - (c_e - c_{e,0}). \tag{16}$$

Here, $\Delta$ and $\Delta_s$ denote the Laplacian operator in three-dimensional bulk space and the Laplace-Bertrami operator on the two-dimensional surface, respectively. The boundary conditions of **Equation 12** and **Equation 13** are

$$-D\nabla_{\mathbf{n}} c_D = c_D(\omega_D + \omega_{dD} c_d)(1 - c_d - c_{de}) - c_{de} \tag{17}$$

$$-D\nabla_{\mathbf{n}} c_E = \omega_E c_E c_d - (c_e - c_{e,0}). \tag{18}$$

Here, $\nabla_{\mathbf{n}}$ is the derivative along the normal direction to the membrane. In this model, the set of concentration fields is expressed by $\Psi = (c_d, c_{de}, c_e, c_D, c_E)$, where the membrane concentration fields are $\psi = (c_d, c_{de}, c_e)$ and the bulk concentration fields are $\phi = (c_D, c_E)$.

## Appendix 3

DOI: https://doi.org/10.7554/eLife.44591.033

## Theoretical model II

The effect of ATP hydrolization of MinD in bulk plays an essential role in the model proposed by *Huang et al. (2003)*. This model assumes that MinE is in the complex form of MinDE on the membrane. In a previous report (*Denk et al., 2018*), the model is generalized to include the effect of the formation of the MinDE complex from membrane-bound MinD and MinE. We, therefore, considered the following model:

Model II

$$\partial_t(c_{DT} + c_{DD}) = D\Delta(c_{DT} + c_{DD}) \tag{19}$$

$$\partial_t c_{DD} = D\left(\Delta - \frac{1}{\xi^2}\right)c_{DD} \tag{20}$$

$$\partial_t c_E = D\Delta c_E \tag{21}$$

$$\partial_t c_d = \Delta_s c_d + c_{DT}(\omega_D + \omega_{dD}c_d) - \omega_E c_E c_d - \omega_{ed}c_e c_d \tag{22}$$

$$\partial_t c_{de} = \Delta_s c_{de} + \omega_E c_E c_d + \omega_{ed}c_e c_d - c_{de} \tag{23}$$

$$\partial_t c_e = \Delta_s c_e + c_{de} - \omega_{ed}c_e c_d - \left(c_e - c_{e,0}\right). \tag{24}$$

The length scale associated with ATP hydrolization is denoted by $\xi = \sqrt{D/\lambda}$, where $\lambda$ is the rate of ATP hydrolization. The boundary conditions of *Equations 19 and 21* are:

$$-D\nabla_{\mathbf{n}}(c_{DT} + c_{DD}) = c_{DT}(\omega_D + \omega_{dD}c_d) - c_{de} \tag{25}$$

$$-D\nabla_{\mathbf{n}}c_{DD} = -c_{de} \tag{26}$$

$$-D\nabla_{\mathbf{n}}c_E = \omega_E c_E c_d - \left(c_e - c_{e,0}\right). \tag{27}$$

Chemical reactions are schematically shown in *Figure 5—figure supplement 1B*. Except for ATP-hydrolization, this model differs from *Equations 12 and 16* only in the saturation of the membrane-bound MinD in *Equations 22 and 25*. As we show in the analysis in *Figure 6B*, this effect is not relevant in a closed membrane. In fact, when $\xi \geq R$, this model reproduces waves that are similar to those in the model I, whereas when $\xi << R$ and $\omega_{ed} << 1$, this model reproduces a standing wave similar to that from the initial condition in which $c_d$ accumulated in a semi-sphere on the membrane (*Huang et al., 2003*; *Fange and Elf, 2006*). We use the same parameters as Model I, and the additional parameter is set to be $\lambda = 1.0$. In this model, it is convenient to choose the set of concentration fields to be expressed by $\Psi = (c_d, c_{de}, c_e, c_{DT} + c_{DD}, c_{DD}, c_E)$, where the membrane concentration fields are $\psi = (c_d, c_{de}, c_e)$ and the bulk concentration fields are $\phi = (c_{DT} + c_{DD}, c_{DD}, c_E)$.

## Saturation of membrane-bound proteins does not play a role in the closed membrane

Model I differs from Model II in two respects. One is ATP hydrolysis in bulk proposed by *Huang et al. (2003)*. The second effect is saturation membrane-bound MinD, that is, the concentration of MinD does not exceed a certain value (one in our unit) which is given as a phenomenological parameter. This term was questioned by *Halatek and Frey (2014)*, in

which ATP hydrolysis in bulk caps the concentration without this term. In order to show that the saturation term is not necessary in a small system, even without ATP hydrolysis in bulk, we compare the stability analyses of Model I with and without the saturation term (*Figure 6B* and *Figure 6—figure supplement 1*). The results show that they are almost identical, and the same mechanism of wave instability, namely suppression of instability at the zero mode, occurs in both cases. This is because maximum concentration of MinD on the membrane is not set by the saturation term but rather by conservation law. For a larger system, this is not the case because the bulk concentrations is insensitive to the membrane concentrations due to small $\alpha$.

## Appendix 4

DOI: https://doi.org/10.7554/eLife.44591.033

# Generic model and its reduction onto membrane

In order to give a unified expression for different models (Model I and II), we investigated a generic form of these models. We considered $n$ variables of membrane-bound proteins denoted by $\psi$ such as $\psi = (c_d, c_{de}, \cdots)$, and $m$ variables of bulk cytosol concentrations denoted by $\phi$ such as $\phi = (c_{DD}, c_E, \cdots)$. The dynamics in bulk are expressed by linear equations such as:

$$\partial_t \phi_i = (D\Delta - \lambda_i)\phi_i. \tag{28}$$

The subscript denoted a specific concentration of a protein in the bulk cytosol. The dynamics of the membrane concentrations is formally written as:

$$\partial_t \psi_i = F_i \left[ \{\psi_p\}_{p \in [1,n]}, \{\phi_q\}_{q \in [1,m]} \right] + \Delta_s \psi_i, \tag{29}$$

where the first term on the right-hand side expresses biochemical reactions. The boundary conditions are, in general, nonlinear, but they are rewritten as:

$$D\nabla_{\mathbf{n}} \phi_i = p_{ij}\psi_j + q_{ij}\partial_t\psi_j - r_{ij}\Delta_s\psi_j. \tag{30}$$

The matrices, $\mathbf{p}$, $\mathbf{q}$, and $\mathbf{r}$, are specified by each model. A homogeneous stationary solution is obtain by $F_i[\psi^*, \phi^*] = 0$ together with conservation law of MinD and MinE.

The concentration fields in the linearized equation are expanded with spherical harmonics (*Equation 8*) for the closed membrane or with wave vectors $\mathbf{k}$ (*Equation 10*) for the planar membrane. Using the eigenvalues, the concentration fields are expressed as $\psi(t) = \psi^* + \delta\psi e^{\sigma t}$ on the membrane and as $\phi(t) = \phi^* + \delta\phi e^{\sigma t}$ in bulk. We can solve *Equation 28* together with the boundary conditions (*Equation 30*). Then the linearized equation on the membrane is expressed as:

$$\sigma\delta\psi_{lm,i} = \sum_{j=1}^{n} \Lambda_{l,ij}^{(0)}\delta\psi_{lm,j} + \sum_{j=1}^{m} \Gamma_{ij}\delta\phi_{lm,j}(R), \tag{31}$$

where the $n \times n$ matrix $\Lambda_l^{(0)}$ is expressed by

$$\Lambda_{l,ij}^{(0)} = \frac{\partial F_i}{\partial\psi_j}\Big|_{\psi=\psi^*, \phi=\phi^*} - \frac{l(l+1)}{R^2}\delta_{ij}, \tag{32}$$

and the coupling between membrane and bulk dynamics is expressed by $n \times m$ matrix $\Gamma$ as

$$\Gamma_{ij} = \frac{\partial F_i}{\partial\phi_j}\Big|_{\psi=\psi^*, \phi=\phi^*}. \tag{33}$$

For the planar membrane, the wave-number-dependent concentrations $\phi_k$ and $\psi_k$ were considered, and $\Lambda_l^{(0)}$ and $l(l+1)/R^2$ were replaced by $\Lambda_k^{(0)}$ and $k^2$, respectively. To obtain the eigenvalues, we solved the equation in which the determinant of the linear matrix in *Equation 31* vanished (see *Equation 1*). This is similar to *Equation 11*, but it has a $(n+m) \times (n+m)$ matrix. The current form has only a $n \times n$ matrix, which makes the effect of confinement clearer, as shown in *Equation 1*.

Because there are two conserved quantities of this system, total MinD and MinE concentrations, there are two zero eigenvalues (zero eigenmodes) associated with them. Except for these eigenvalues, the bottom-right block of the matrix $\Lambda_l$ associated with bulk concentrations is invertible, and thus, we may eliminate the bulk concentrations by solving the linearized equations. This argument assumes that wave instability does not occur by the eigenvalues at the finite modes connected to the zero eigenmodes. This is the case in the models studied here, and the results in *Halatek and Frey (2018)* for the planar membrane

with the model proposed by *Huang et al. (2003)* also demonstrate that the instability at a finite wave number is not connected to the zero eigenmodes.

