## [Decision Letter]

[Editors’ note: the authors were asked to provide a plan for revisions before the editors issued a final decision. What follows is the editors’ letter requesting such plan.]

Thank you for sending your article entitled "Cell-sized confinement controls generation and stability of a protein wave for spatiotemporal regulation in cells" for peer review at *eLife*. Your article is being evaluated by three peer reviewers, and the evaluation is being overseen by a Reviewing Editor and Arup Chakraborty as the Senior Editor.

Given the list of essential revisions, including new experiments, the editors and reviewers invite you to respond within the next two weeks with an action plan and timetable for the completion of the additional work. We plan to share your responses with the reviewers and then issue a binding recommendation.

As you will see from the complete reviews below, there are two very major issues that the referees have identified, and these have been discussed extensively between the reviewers and the Reviewing Editor.

The first is the attention given to the scientific writing. The second concerns the interpretation of the effects of BSA. To rule out that the observations are an artefact, the reviewers would like to see an experiment with labelled BSA. If it remains cytosolic at low concentrations, we could be convinced that the effect is occurring in solution. If it accumulates at the membrane, it is a sign of membrane failure.

To investigate whether this "crowding" effect is specific to BSA, we further suggest experiments with an alternative protein crowder, preferably with a label so it can be followed. As the manuscript already showed measurements with sfGFP, that could simply be used in high concentrations.

We would emphasize that should the effect be specific to BSA we are not requesting a full investigation of the effect.

*Reviewer #1:*

In this work, the authors present a combined experimental (in vitro) and theoretical study of the pattern formation by the Min proteins of *E. coli*. Due to its simplicity and its beauty, the system has become one of the paradigmatic systems for studying protein self-organisation in cells. In spite of 20 years of research, the mechanism underlying Min-protein pattern formation is still not understood in detail. Notably, the relation between patterns observed in vivo and in vitro remains unclear. Here, the authors study the Min protein dynamics in spherical vesicles made of *E. coli* lipids in presence or absence of crowding agents. They find that the lipid composition affects the conditions for spontaneous wave emergence. Addition of BSA as a crowding agent altered conditions for the appearance of Min-protein waves. Other crowding agents (PEG8000, Ficoll70) failed to do so. These results were obtained for conditions that lead to the spontaneous emergence of traveling Min-protein waves on flat supported lipid bilayers. The authors then go on to reveal the origin of this effect and find that it depends changes of MinE's binding rate to the membrane. The authors infer this from the ratio of the amount of MinE bound to the membrane to that in the cytoplasm. The results of the theoretical analysis are in line with these conclusions.

I find the results interesting and think that they make a valuable contribution to our understanding of the Min system. Still I do not think that the manuscript can be published in its current state. Most importantly – I am sorry to be so blunt – it is not well written. A good impression can be gained by reading the Abstract. It is very difficult to understand and does not clearly convey the message of the work. Unfortunately, this is true for essentially the whole work. In addition to often being hard to understand, the text is also too vague. To give but one example: "To quantify the details of MinE localization in microdroplets, we employed an index value for localization ratio (c/m) obtained by dividing quantities of MinE in cytosolic parts (c) by those on membranes (m)." (ll. 178) How are these quantities measured? What are their units? Also the definition of the computational models does not become clear from the main text.

Finally, in my opinion, the theoretical results should take more weight in the main text.

In conclusion, I think that very thorough revisions of the manuscript are necessary to make this work acceptable for publication in *eLife*. I also think that the message would be much stronger if the authors could measure the rates of MinE membrane binding, but I acknowledge that this would require a lot more work that goes beyond what the authors aimed for here.

*Reviewer #2:*

Kohyama et al. generate water-in-oil emulsion droplets to encapsulate MinDE and observe pattern formation. They show that when using *E. coli* polar lipids to generate the droplets a high concentration of BSA is necessary to sustain pattern formation of MinDE. They attribute this to a crowding effect of the BSA. However, other crowding agents such as PEG or Ficoll do not have the same effect. Using a previously published protocol that uses DOPC and Cardolipin for droplet formation, the authors can also generate droplets that support pattern formation in the absence of BSA. The authors claim that the BSA changes the ability of MinE to localize to the membrane and quantify this effect under different conditions using c/m ratio (cytosolic to membrane localization of MinE).

We believe that the presented work is methodologically rigorous, but does not represent a significant advance to justify publication in *eLife*. The manuscript reproduces MinDE pattern formation in micro-sized compartments. There have been several such studies in compartments lined with SLBs (Zieske et al., 2013; Zieske et al., *eLife* 2014; Caspi and Dekker, 2016) and in droplets (Zieske et al., 2016) and in vesicles (Litschel et al., 2018).

Furthermore, we are concerned with the interpretation of the major results in the manuscript: The authors claim that crowding through BSA prevents MinE binding to the droplet interface which they show is essential for emergence of Min patterns (Figure 2D). There are several problems with this hypothesis. First, as the authors themselves show, other crowding agents like PEG8000 and Ficol70, both commonly used crowders, cannot reproduce the effect obtained with BSA, namely the occurrence of Min patterns (Figure 3D). Second, the authors show that BSA in their assay does not work as a crowder, as they show that the diffusion in solution of in this case sfGFP is only impacted by BSA at amounts higher than 100 mg/ml (Figure 2A), but MinDE reliably form patterns even at 10 mg/ml BSA (subsection “Protein crowding inhibits spontaneous localization of MinE on membranes” and “Space sizes of microdroplets changes rates of spontaneous MinE membrane localization”). Third, the authors refer to and reproduce an older publication (Zieske et al., 2016) where it has been shown that MinDE patterns emerge in droplets formed with DOPC and Cardiolipin even in the absence of BSA (Video S2). The authors do not explain why in this case MinDE can form patterns in a microsized compartment without crowding other than sloppily referring to difference in lipids (subsection “MinDE are insufficient for emerging Min waves in micro-sized space fully covered with *E. coli* polar lipid”).

We believe that what the authors are looking at is that BSA patches the faulty monolayer. BSA is regularly used to passivate artificial membranes. The low amount of BSA that does not induce crowding but would efficiently passivate the monolayer interface speaks for such an interpretation of the results. As MinD in the presence of ATP always binds to the interface, we propose to use an altogether different protein, such as a soluble fluorescent protein (which the authors seemed to have done already as they show diffusion data for sfGFP), as a test for membrane integrity. Another control would be to use labelled BSA to see if it actually patches the membrane or does indeed stay in solution and act as a crowder as suggested by the authors.

*Reviewer #3:*

The manuscript by Kohyama et al. reports a study of the impact of cell-sized confinement on Min-wave generation and stability. The study is timely as the studies on Min system in vitro were mostly done on two-dimensional planar membranes, whereas the Min system in vivo operates in a finite volume enclosed by the cell membrane. Indeed, given the very different surface area to volume ratio of 2D planar membranes compared to micron-sized cells, it is unclear if the behavior of the Min systems in the former geometry has anything to do with the behavior in the latter. To remedy this situation, the authors used an emulsification method to encapsulate MinD, MinE, and ATP in micron-sized vesicles covered with *E. coli* polar lipids. The authors found that encapsulation did not lead to Min waves, despite the use of physiological levels of MinE, MinD, and ATP. However, the authors noted that encapsulation resulted in a low ratio of "cytosolic" (i.e. bulk 3D) MinE to membrane-associated MinE, and hypothesized that this low c/m ratio could be responsible for suppression of Min waves. The authors then showed that the c/m ratio could be dramatically raised by addition of a protein-based crowding agent BSA and this was sufficient to induce Min waves. While the role of BSA could be via crowding, addition of alternative crowding agents (PEG8000 or Ficoll70) did not induce oscillations or suppress MinE membrane association. To complement their experimental observations, the authors modeled the reaction-diffusion Min system in a confined geometry, and showed that the degree of spontaneous binding of MinE to the membrane shifts the conditions for Min wave emergence. Using computational simulations, the authors further explored the difference in Min wave generation mechanism on 2D planar membrane and in 3D micron-sized spaces, and highlighted the effect of cell-size confinement in reducing the parameter ranges for Min wave emergence.

Overall, the dependence of MinE membrane localization and the emergence of Min waves on physiological levels of protein-based crowders reported in this study could be an exciting contribution to the literature, provided that the authors satisfactorily address the issues described below:

1) The authors' finding that BSA induces Min waves by suppression of the spontaneous membrane binding of MinE is interesting. However, the manuscript is not clear about how BSA suppresses the spontaneous membrane binding of MinE. Is it a specific property of BSA, or a general property of protein-based crowding agents? If the effect is specific to BSA, then it is unclear how this observation relates to in vivo Min waves, and the biological significance of the observations with BSA seems doubtful. The biological relevance of the work would be more convincing if the authors can demonstrate that Min waves can be induced by physiological levels of at least one other type of protein crowding agent.

2) There is a disconnect between the experiment and modeling. In the experiment, the authors showed that the cell-sized confinement results in a low c/m ratio of MinE, but that this effect can be "canceled" by using BSA. Therefore, the study suggests a potential role of the bacterial cytoplasm as a key component of the Min system that suppresses the spontaneous membrane binding of MinE, and allows the emergence of Min waves. However, toward the end of modeling, the authors appear to state that a cell-size 3D space narrows the tolerant range of conditions for Min wave emergence, and only physiologically relevant concentrations of MinD and MinE can induce Min waves in a cell-sized space (apparently without using BSA, though this is not clearly stated). This is confusing, as if the cytoplasm acts like BSA, then according to the experiments with BSA one should see in vivo Min waves over a wide range of MinE and MinD levels, but this seems not to be the case for in vivo systems (though the authors do not describe the dependence of in vivo Min waves on MinD and MinE levels, and this is an omission). This results of the modeling analysis are confusing, as if the cytoplasm is indeed canceling the confinement effect then in principle it should allow a wider ranges of conditions for Min wave emergence. The authors need to clarify the logic of their conclusions from modeling.

[Editors' note: further revisions were requested prior to acceptance, as described below.]

Thank you for resubmitting your work entitled "Cell-sized confinement controls generation and stability of a protein wave for spatiotemporal regulation in cells" for further consideration at *eLife*. Your revised article has been favorably evaluated by Arup Chakraborty (Senior Editor), a Reviewing Editor, and three reviewers.

The manuscript has been improved but there are some remaining issues that need to be addressed before acceptance, as outlined below. We include lightly edited comments from each of the reviewers for clarity regarding how you might make changes to the current manuscript.

*Reviewer #1:*

I appreciate the effort done by the authors to improve the writing and to clarify their results. It now becomes clear that BSA suppresses spontaneous membrane-binding of MinE, that is, in absence of MinD. It is a nice result that the ratio c/m between MinE in the cytoplasm/buffer and on the membrane is an important control parameter. (The mechanism underlying the suppression of MinE membrane-binding remains unknown.) It notably sheds light on the topic of the work, namely the impact of (3d) geometry on Min-protein self-organisation. – The computational part has also improved. It contains a number of interesting observations on the mechanism of wave generation. I did not quite understand what the discussion about excitability adds to our understanding of the role of geometry on Min-protein self-organization.

In spite of the clear improvements of the text, there are still statements that are unclear to me. I have listed a few below. Also, I do not understand what Appendix 1 contributes.

In conclusion, the authors manage to clarify differences between the Min dynamics in closed and open geometries and thereby show that the mechanism for Min protein self-organization in vitro and in vivo are fundamentally the same – this was disputed by Caspi and Dekker, *eLife* (2016). In my opinion this result can be published in *eLife*. Still, I would suggest that the writing be further improved.

Some specific points:

Subsection “BSA modifies attachment of MinE on membranes without MinD”; "Thus, we first assumed that BSA may change reaction rates of MinD and MinE binding each other or to the membrane, decrease diffusion constants of MinD and MinE in the cytosol, and increase association rates between macromolecules by a depletion force. Among these effects, the effect of depletion force is excluded […]" I am not sure what the authors want to say here. They first made some assumptions; and then what? Is only the last effect due to depletion forces and thus only this one excluded? The following sentence and paragraph starts with another "first". I am confused here.

In the same section: "it was found that even 100 mg/mL of BSA slightly but did not significantly decrease the diffusion rates of MinD" Sounds awkward.

In subsection “Suppression of spontaneous membrane localization of MinE is the key to emergence of Min waves in micro-sized space” paragraph four: "(Figure 4D)." Should be (Figure 3D) and (Figure 4E)." Should be (Figure 3E)

Subsection “Space sizes of microdroplets changes the rate of spontaneous MinE membrane localization”: "As spatial factors to determine c/m of MinE, we can raise maximum levels of attachment on membranes and total amounts of MinE" I do not understand this sentence. Why is the total amount a spatial factor?

The second paragraph in subsection “Space sizes of microdroplets changes the rate of spontaneous MinE membrane localization” is altogether quite unclear. Some specific example to underline my statement:

"In small microdroplets with 10 μm diameter, c/m was near 0 and the value (c/m) increased in higher concentrations of MinE. In larger microdroplets, c/m increased in proportion to the MinE amounts." What is the difference between the small and larger microdroplets?

"(…) c/m was maintained low within <130 μm (…)" ?

"(…) in the presence of 10 mg/mL BSA, a minimum BSA concentration for Min wave emergence in microdroplets, c /m did not increase in size but was maintained high irrespective of droplet size" ?

"(…) at high concentration of MinE, Min wave did not emerge while MinE indicated high c/m ratio" ?

Subsection “Computational simulation for Min wave supports the importance of MinE localization for wave emergence”: "c_e_ = 0 when D_0_ = 0" these quantities have not been introduced.

*Reviewer #2:*

The overall quality of the manuscript has been clearly improved. However, there are several logical contradictions that remain to be addressed before publication. Still, the main effect the authors describe (and also theoretically discuss) seems that BSA outcompetes MinE for membrane and/or interface binding, thereby allowing for pattern formation under sub-optimal conditions. The experimental results are rather specific, but could be interesting from a technical point of view.

General concerns and contradictory logic:

Overall, the writing of the manuscript is much better. However, there remain several logical contradictions that need to be resolved before publication:

The main result of the study is that proteinaceous crowders such as BSA or *E. coli* extract can suppress MinE binding to the droplet interface and therefore induce Min wave emergence even with lipid compositions that usually do not support pattern formation in droplets. In contrast, addition of other crowders such as PEG8000 does not produce the same effect. However, using high MinE concentration itself or previously published lipid formulations, the authors could also obtain droplets with wave formation. The authors claim that BSA could suppress MinE membrane binding by either of two mechanisms:

1) BSA/extract could directly bind to the membrane (as reported previously and cited by the authors) and thereby competitively inhibit membrane binding of MinE

2) BSA/extract could regulate the conformational switching of MinE by excluded volume effects. Thereby reducing MinE membrane attachment, as it would supposedly remain longer in closed conformation

The authors contradict both of these explanations with experiments or in the main text.

Re explanation 1): They performed the additional experiment requested in review, showing that neither labelled BSA nor sfGFP-MinD seem to bind to the membrane (Figure 2D). However, the setup of the experiment is problematic. The authors used 1 µM labelled BSA together with 1 mg/ml non-labeled BSA (about 16 µM). It could be that the signal of membrane-bound BSA is therefore too weak to be detected. To disentangle this, we would suggest the authors use different concentration of purely labelled BSA, starting with low concentrations.

Re explanation 2): The authors cite in the main text that BSA shows little depletion force compared to PEG8000 or Ficoll70. However, these polymer crowders do not support pattern formation, contradicting the statement that the depletion force is likely to be causing MinE conformational switching.

The third explanation not mentioned by the authors could still be that BSA passivates the water oil interface, preventing MinE attachment and denaturation. This effect is hard to access experimentally, because it could also take place during droplet formation, when lipids and proteins come to the interface. If the lipid monolayer is not generated fast enough, proteins could denature, but a proper monolayer could still form. Hence, it would be more convincing if the FRAP data on the protein in presence and absence of BSA would not be presented for MinD, but for MinE.

Furthermore, in the main text they demonstrate that I74M has a lower c/m ratio than WT MinE and is less influenced by BSA. They say that congruently this mutant does not support pattern formation. However, in the discussion they rightfully point out that this mutant does not show pattern formation under any of their conditions (the ones they tested) and also not in vivo. We still believe that this experiment needs to be removed in the current form, unless the authors can demonstrate that the mutant is able to form patterns in the droplet under some condition (e.g. at lower MinE concentration as demonstrated in vitro using the better characterized open MinE mutant I24N, see Denk et al., 2018) or by using the characterized MinE mutant I24N in the first place. The authors then need to show that the conditions for wave emergence are differentially influenced by BSA addition.

---

## [Author Response]

[Editors' note: the authors’ plan for revisions was approved and the authors made a formal revised submission.]

As you will see from the complete reviews below, there are two very major issues that the referees have identified, and these have been discussed extensively between the reviewers and the Reviewing Editor.The first is the attention given to the scientific writing. The second concerns the interpretation of the effects of BSA. To rule out that the observations are an artefact, the reviewers would like to see an experiment with labelled BSA. If it remains cytosolic at low concentrations, we could be convinced that the effect is occurring in solution. If it accumulates at the membrane, it is a sign of membrane failure.To investigate whether this "crowding" effect is specific to BSA, we further suggest experiments with an alternative protein crowder, preferably with a label so it can be followed. As the manuscript already showed measurements with sfGFP, that could simply be used in high concentrations.We would emphasize that should the effect be specific to BSA we are not requesting a full investigation of the effect.

i) Problems on scientific writing

We agree to the reviewer comments that scientific writing in our original manuscript was vague and unfocused. We totally revised our manuscript to fix the insufficient explanations and to clarify the logical flow.

Detailed explanation:

i-a) We deeply appreciate that the reviewers pointed out typos, erroneous/insufficient descriptions, citation problems, and misleading sentences. We carefully revised them following the reviewers’ comments.

i-b) We are unintentionally biased so that essentially all the details on modeling and numerical simulations are placed in the Supplementary Materials. This placing makes the manuscript harder to read without reading all the details in Supplementary Materials. Details of the simulation have been moved to Materials and methods and Appendix in the main text. Furthermore, we have added a description of the results by the numerical simulations and theoretical analysis in main text.

i-c) We agree that there are several unclear logical flows in our previous manuscript. We revised them by rephrasing sentences, the addition of a description for the simulation, and division and reconstruction of sections and figures for readability.

i-d) For further readability, our manuscript was edited by commercial a English proofreading service.

ii) Interpretation of the effects of BSA

We clarified the interpretation of the effects of BSA by additional experiments through two approaches. First, we showed that our results were not due to an artefact of our experimental system. Second, following reviewer #3’s comment “The biological relevance of the work would be more convincing if the authors can demonstrate that Min waves can be induced by physiological levels of at least one other type of protein crowding agent”, we added new data on Min wave generation using *E. coli* cell extract, which is a more physiological crowder.

Detailed explanation:

ii-a) Membrane integrity of our experimental system

To check the membrane integrity concern raised by reviewer #2, we have examined the localization assay using labeled-BSA and sfGFP alone, respectively. As a result, we observed cytosolic localization of labeled BSA and sfGFP, and therefore, we believe there is no sign of membrane failure in our experimental system.

ii-b) Min wave emergence by another crowding agent

We tested the effect of another crowding agent, *E. coli* cell extract, which is used in other studies as similar protein crowder to BSA (Groen et al., 2015). In agreement with our early results, we found that *E. coli* cell extract can work as BSA to generate Min wave in lipid microdroplets and to suppress the MinE localization on the membrane. We believe these results make our conclusion clearer and stronger.

Reviewer #1:[…] I find the results interesting and think that they make a valuable contribution to our understanding of the Min system. Still I do not think that the manuscript can be published in its current state. Most importantly – I am sorry to be so blunt – it is not well written. A good impression can be gained by reading the Abstract. It is very difficult to understand and does not clearly convey the message of the work. Unfortunately, this is true for essentially the whole work. In addition to often being hard to understand, the text is also too vague. To give but one example: "To quantify the details of MinE localization in microdroplets, we employed an index value for localization ratio (c/m) obtained by dividing quantities of MinE in cytosolic parts (c) by those on membranes (m)." (ll. 178) How are these quantities measured? What are their units? Also the definition of the computational models does not become clear from the main text.

We appreciate that the reviewer pointed out our poor writing and vague statements. We carefully checked our manuscript again, and rewrote it in order to make a logical flow clearer. For this purpose, necessary descriptions including methods for c/m quantification were added in the main text. Furthermore, our manuscript is now edited by commercial editing service.

Finally, in my opinion, the theoretical results should take more weight in the main text.

Descriptions of simulation results were added in the main text to take more weight.

In conclusion, I think that very thorough revisions of the manuscript are necessary to make this work acceptable for publication in eLife.

According to the comment, our manuscript is now totally revised as described above.

I also think that the message would be much stronger if the authors could measure the rates of MinE membrane binding, but I acknowledge that this would require a lot more work that goes beyond what the authors aimed for here.

Although the rate of MinE membrane binding is very important parameter, it is difficult to evaluate membrane binding rates of MinE due to technical difficulty in evaluation those in lipid microdroplets. This point is now described in Discussion.

Reviewer #2:[…] We believe that the presented work is methodologically rigorous, but does not represent a significant advance to justify publication in eLife. The manuscript reproduces MinDE pattern formation in micro-sized compartments. There have been several such studies in compartments lined with SLBs (Zieske et al., 2013; Zieske et al., eLife 2014; Caspi and Dekker, 2016) and in droplets (Zieske et al., 2016) and in vesicles (Litschel et al., 2018).

We respectfully disagree our present study is merely a reproduction of the previous work because their studies analyze the patterns of Min waves but did not mention the conditions for Min wave emergence in micro-sized compartment which is a main point of our present study. Here we demonstrate that the conditions for Min wave emergence in closed geometry differ from those in open spaces, and revealed the reason by experiments and theoretical analysis. We believe these findings bring significant advances and justify publication of our present study in *eLife*.

Furthermore, we are concerned with the interpretation of the major results in the manuscript: The authors claim that crowding through BSA prevents MinE binding to the droplet interface which they show is essential for emergence of Min patterns (Figure 2D). There are several problems with this hypothesis. First, as the authors themselves show, other crowding agents like PEG8000 and Ficol70, both commonly used crowders, cannot reproduce the effect obtained with BSA, namely the occurrence of Min patterns (Figure 3D).

As mentioned above, we showed that the effect of BSA to generate Min waves in cell-sized space is suppression of spontaneous localization of MinE on membrane (denoted as c/m in experiments and *c_e,0_* in numerical simulation) rather than simple crowding effects such as limitation of diffusion. While BSA and cell extract suppress the spontaneous MinE localization, the non-protein crowders do not show this (Figure 3F). This point is now described in the Discussion.

Second, the authors show that BSA in their assay does not work as a crowder, as they show that the diffusion in solution of in this case sfGFP is only impacted by BSA at amounts higher than 100 mg/ml (Figure 2A), but MinDE reliably form patterns even at 10 mg/ml BSA (subsection “Protein crowding inhibits spontaneous localization of MinE on membranes” and “Space sizes of microdroplets changes rates of spontaneous MinE membrane localization”).

This issue depends on the definition of crowder. The reviewer is correct if the crowder plays a role in suppression of diffusion. However, our present study demonstrated the role of BSA is not the suppression of diffusion but suppression of MinE localization. Nevertheless, we agree that the term “crowding” is confusing to potential readers. We totally revised the Discussion to clarify our conclusion, and stated that the role of BSA and cell extract is not derived from limitation of diffusion.

Third, the authors refer to and reproduce an older publication (Zieske et al., 2016) where it has been shown that MinDE patterns emerge in droplets formed with DOPC and Cardiolipin even in the absence of BSA (Video 2). The authors do not explain why in this case MinDE can form patterns in a microsized compartment without crowding other than sloppily referring to difference in lipids (subsection “MinDE are insufficient for emerging Min waves in micro-sized space fully covered with E. coli polar lipid”).

We are afraid of that the issue is due to the bad scientific writing as pointed out by other reviewers. We explained the reason that MinDE patterns emerge in droplets formed with DOPC and Cardiolipin even in the absence of BSA is suppression of MinE localization due to low amounts of anionic lipids. We have rephrased the corresponding explanations and mentioned these points in the Discussion.

We believe that what the authors are looking at is that BSA patches the faulty monolayer. BSA is regularly used to passivate artificial membranes. The low amount of BSA that does not induce crowding but would efficiently passivate the monolayer interface speaks for such an interpretation of the results. As MinD in the presence of ATP always binds to the interface, we propose to use an altogether different protein, such as a soluble fluorescent protein (which the authors seemed to have done already as they show diffusion data for sfGFP), as a test for membrane integrity. Another control would be to use labelled BSA to see if it actually patches the membrane or does indeed stay in solution and act as a crowder as suggested by the authors.

To check the membrane integrity concern raised by the reviewer, we have examined the localization assay using labeled-BSA and sfGFP alone, respectively. As a result, we observed cytosolic localization of labeled BSA and sfGFP (Figure 2D), and therefore, we believe there is no sign of membrane failure in our experimental system.

Reviewer #3:[…] Overall, the dependence of MinE membrane localization and the emergence of Min waves on physiological levels of protein-based crowders reported in this study could be an exciting contribution to the literature, provided that the authors satisfactorily address the issues described below:1) The authors' finding that BSA induces Min waves by suppression of the spontaneous membrane binding of MinE is interesting. However, the manuscript is not clear about how BSA suppresses the spontaneous membrane binding of MinE. Is it a specific property of BSA, or a general property of protein-based crowding agents? If the effect is specific to BSA, then it is unclear how this observation relates to in vivo Min waves, and the biological significance of the observations with BSA seems doubtful. The biological relevance of the work would be more convincing if the authors can demonstrate that Min waves can be induced by physiological levels of at least one other type of protein crowding agent.

Following to the reviewer’s comment, we added new data on c/m change and Min wave generation as a result of supplementation with *E. coli* cell extract, which is a more physiological crowder. As a result, cell extract also suppressed spontaneous membrane localization of MinE and generated Min waves in microdroplets. Furthermore, the behavior against concentration shift of cell extract is similar to the case for BSA. We believe these results strengthened our conclusion.

2) There is a disconnect between the experiment and modeling. In the experiment, the authors showed that the cell-sized confinement results in a low c/m ratio of MinE, but that this effect can be "canceled" by using BSA. Therefore, the study suggests a potential role of the bacterial cytoplasm as a key component of the Min system that suppresses the spontaneous membrane binding of MinE, and allows the emergence of Min waves. However, toward the end of modeling, the authors appear to state that a cell-size 3D space narrows the tolerant range of conditions for Min wave emergence, and only physiologically relevant concentrations of MinD and MinE can induce Min waves in a cell-sized space (apparently without using BSA, though this is not clearly stated). This is confusing, as if the cytoplasm acts like BSA, then according to the experiments with BSA one should see in vivo Min waves over a wide range of MinE and MinD levels, but this seems not to be the case for in vivo systems (though the authors do not describe the dependence of in vivo Min waves on MinD and MinE levels, and this is an omission). This results of the modeling analysis are confusing, as if the cytoplasm is indeed canceling the confinement effect then in principle it should allow a wider ranges of conditions for Min wave emergence. The authors need to clarify the logic of their conclusions from modeling.

Our modeling section had two parts: one is MinE localization discussed above and the other is the effect of confinement. We have rewritten the main text so that we have separated these sections and clarified the logic. To clarify the relation between c/m and theoretical analysis, we have revised the figure (Figure 5) and rewritten the modeling section entirely. Although the model cannot say anything about molecular mechanism of MinE delocalization due to BSA, we have clarified increasing c/m (decreasing *c_e,0_*) enhances the Min wave generation. Our new experiment uses cell extract to reproduce the wave, and therefore, it supports the idea that BSA behaves as cytosol in the sense that it cancels MinE localization on the membrane.

The section for the effect of confinement explains the reason why the planar membrane is robust against MinE localization. This section shows findings that are independent from the results form spontaneous MinE localization. To make this point clearer, we divided figures and sections.

[Editors' note: further revisions were requested prior to acceptance, as described below.]

Reviewer #1:I appreciate the effort done by the authors to improve the writing and to clarify their results. It now becomes clear that BSA suppresses spontaneous membrane-binding of MinE, that is, in absence of MinD. It is a nice result that the ratio c/m between MinE in the cytoplasm/buffer and on the membrane is an important control parameter. (The mechanism underlying the suppression of MinE membrane-binding remains unknown.) It notably sheds light on the topic of the work, namely the impact of (3d) geometry on Min-protein self-organisation. – The computational part has also improved. It contains a number of interesting observations on the mechanism of wave generation. I did not quite understand what the discussion about excitability adds to our understanding of the role of geometry on Min-protein self-organization.

We appreciate the comments by the reviewer. We proposed excitability as a second possibility of robustness of the Min wave generation in open geometry. It suggests that the wave may occur even when the stationary state is linearly stable due to large perturbation (as shown in Figure 5 that the blue points are positive even away from the region of linear instability on a planar membrane). To show excitability, a large amount of Min proteins necessarily attaches to the membrane in a local space as shown Figure 8A and B. This is possible only in the open geometry, and not in the closed membrane because the process strongly modifies the bulk concentrations due to the coupling between the bulk and membrane concentrations. To clarify this issue, we added the corresponding discussion as follows:

“We stress that excitability occurs only in the flat membrane because in a closed small membrane, any large change in the local membrane concentrations during an excitable wave inevitably changes the bulk concentrations.”

We also note that excitability suggests that the wave generation may be strongly dependent on an initial condition as described in Discussion. To our knowledge, such dependence is unexplored in the case of Min waves.

In spite of the clear improvements of the text, there are still statements that are unclear to me. I have listed a few below. Also, I do not understand what Appendix 1 contributes.

We also appreciate the comments. In Appendix 1, we summarize our current theoretical understandings on the Min wave generation. The benefit of this description is to compare our theoretical models with previously proposed ones. We have revised Appendix 1 so that it starts from generic phenomenological models to explain what is missing in those models to understand the Min system, and then, we summarized the previous models specific for the Min wave generation.

“Before explaining the theoretical models in this study in Appendix 2 and 3, we summarize current theoretical understandings of the Min wave generation. Spontaneous wave generation observed in the experiments have been studied as wave instability in reaction-diffusion equations.”

Other revisions are described in the responses to the specific pointing by the reviewer.

In conclusion, the authors manage to clarify differences between the Min dynamics in closed and open geometries and thereby show that the mechanism for Min protein self-organization in vitro and in vivo are fundamentally the same – this was disputed by Caspi and Dekker, eLife (2016). In my opinion this result can be published in eLife. Still, I would suggest that the writing be further improved.

We deeply appreciate the reviewer’s comments. The writing has been further improved by rephrasing the paragraphs and sentences pointed by the reviewers, and by a proofreading service again.

Some specific points:Subsection “BSA modifies attachment of MinE on membranes without MinD”; "Thus, we first assumed that BSA may change reaction rates of MinD and MinE binding each other or to the membrane, decrease diffusion constants of MinD and MinE in the cytosol, and increase association rates between macromolecules by a depletion force. Among these effects, the effect of depletion force is excluded […]" I am not sure what the authors want to say here. They first made some assumptions; and then what? Is only the last effect due to depletion forces and thus only this one excluded? The following sentence and paragraph starts with another "first". I am confused here.

When we have considered the effects of BSA, we first speculated the following three effects: BSA induces a depletion force, decreases diffusion constants, and enhances formation of MinDE complexes. However, we found that none of these effects are relevant. Instead, we found that BSA inhibits attachment of MinE on a membrane in the absence of MinD. To clarify what we mean, we revised this paragraph as follows:

“To understand why the condition for the emergence of Min waves differ between being located on 2D planar membranes and in 3D closed geometry, we investigated how BSA affects the mechanism of wave emergence in a closed micro-sized space. Crowding agents, like BSA, may change reaction rates and diffusion rates (23, 24). Changes to the reaction rate are related to changes in the interactions between Min proteins or between a protein and a membrane. The first-known mechanism to modify the interactions is the depletion force, which enhances the attraction between proteins. The crowding agents might also bind directly with MinD or MinE to promote the formation of MinDE complexes. Furthermore, the crowding agents decrease the diffusion constants. We investigated these effects in detail. Among these effects, the effect of depletion force was excluded from the investigation because previous studies have indicated that BSA causes a much weaker depletion force than PEG8000 or Ficoll70 do (24).”

In the same section: "it was found that even 100 mg/mL of BSA slightly but did not significantly decrease the diffusion rates of MinD" Sounds awkward.

The sentence was rephrased as:

“The diffusion rates of MinD on lipid membranes of various sizes of microdroplets decreased only slightly even in the 100 mg/mL of BSA condition (Figure 2B).”

In subsection “Suppression of spontaneous membrane localization of MinE is the key to emergence of Min waves in micro-sized space” paragraph four: "(Figure 4D)." Should be (Figure 3D) and (Figure 4E)." Should be (Figure 3E)

We appreciate this point. We corrected the typo.

Subsection “Space sizes of microdroplets changes the rate of spontaneous MinE membrane localization”: "As spatial factors to determine c/m of MinE, we can raise maximum levels of attachment on membranes and total amounts of MinE" I do not understand this sentence. Why is the total amount a spatial factor?

We appreciate this point. We removed the word “spatial”, because it is unnecessary in this sentence. Furthermore, we rephrased the sentence as follows:

“To determine c/m of MinE, maximum levels of attachment on membranes and the total amounts of MinE are conceivable factors.”

The second paragraph in subsection “Space sizes of microdroplets changes the rate of spontaneous MinE membrane localization” is altogether quite unclear. Some specific example to underline my statement:

To clarify what we mean, we totally revised the paragraph. Responses to the specific points are described below.

"In small microdroplets with 10 μm diameter, c/m was near 0 and the value (c/m) increased in higher concentrations of MinE. In larger microdroplets, c/m increased in proportion to the MinE amounts." What is the difference between the small and larger microdroplets?

To clarify what we mean, we rephrased the sentences as follows:

“In larger microdroplets (>20 μm diameter), c/m increased in proportion to the amount of MinE. Moreover, the increase in c/m was highly dependent on the concentration of MinE.”

"(…) c/m was maintained low within <130 μm (…)" ?

The sentence was rephrases as: “c/m stayed low when the droplet size was less than 130 μm”

"(…) in the presence of 10 mg/mL BSA, a minimum BSA concentration for Min wave emergence in microdroplets, c /m did not increase in size but was maintained high irrespective of droplet size" ?

The sentence were rephrased as follows:

“We also tested the size dependence of c/m in the presence of 10 mg/mL BSA, which is a minimum BSA concentration for Min wave emergence in microdroplets. In this case, c /m did not depend on sizes. However, the value was higher than 0.7”

"(…) at high concentration of MinE, Min wave did not emerge while MinE indicated high c/m ratio" ?

To clarify what we mean, the sentence and its following sentences were rephrased as follows:

“Although c/m is higher than 0.4 in >50 μm microdroplets entrapping 3 μM MinE (Figure 4), no Min wave were observed at 1 μM MinD and 3 μM MinE in the absence of BSA. This result may be associated with the very low rate of Min wave appearance at this concentration ratio, even in the presence of 100 mg/mL BSA (Figure 1E).”

Subsection “Computational simulation for Min wave supports the importance of MinE localization for wave emergence”: "c_e_ = 0 when D_0_ = 0" these quantities have not been introduced.

We appreciate the reviewer’s comment. Explanation of "c_e = 0 when D_0 = 0" has been included in the revised manuscript as follows:

”the concentration of MinE on the membrane, *c*_e_, becomes *c*_e_=0 when the total concentration of MinD, D_0_, is D_0_=0”

Reviewer #2:The overall quality of the manuscript has been clearly improved. However, there are several logical contradictions that remain to be addressed before publication. Still, the main effect the authors describe (and also theoretically discuss) seems that BSA outcompetes MinE for membrane and/or interface binding, thereby allowing for pattern formation under sub-optimal conditions. The experimental results are rather specific, but could be interesting from a technical point of view.

We sincerely appreciate the comments by the reviewer. However, we respectfully disagree our experimental system is under sub-optimal conditions. We selected the lipids and salts to set microdroplet environments to make them more physiological than the elements used in other studies. Furthermore, we showed cell extract, which is a more physiological element, also induces Min wave in the microdroplets we used. Hence, we believe our experimental results that are supported by our theoretical analysis bring significant findings in this field.

General concerns and contradictory logic:Overall, the writing of the manuscript is much better. However, there remain several logical contradictions that need to be resolved before publication:The main result of the study is that proteinaceous crowders such as BSA or E. coli extract can suppress MinE binding to the droplet interface and therefore induce Min wave emergence even with lipid compositions that usually do not support pattern formation in droplets. In contrast, addition of other crowders such as PEG8000 does not produce the same effect. However, using high MinE concentration itself or previously published lipid formulations, the authors could also obtain droplets with wave formation.

We apologize that our poor writing caused a misunderstanding for the reviewer. High MinE concentration does not emerge Min wave in the absence of BSA. To clarify the point, we rephrased the corresponding sentence as follows: (See also the response to the reviewer 1 comment).

“Although c/m is higher than 0.4 in >50 μm microdroplets entrapping 3 μM MinE (Figure 4), no Min wave were observed at 1 μM MinD and 3 μM MinE in the absence of BSA. This result may be associated with the very low rate of Min wave appearance at this concentration ratio, even in the presence of 100 mg/mL BSA (Figure 1E).”

The authors claim that BSA could suppress MinE membrane binding by either of two mechanisms:1) BSA/extract could directly bind to the membrane (as reported previously and cited by the authors) and thereby competitively inhibit membrane binding of MinE2) BSA/extract could regulate the conformational switching of MinE by excluded volume effects. Thereby reducing MinE membrane attachment, as it would supposedly remain longer in closed conformation

Before going into the details of our reply, we should point out that the two mechanisms are possible candidates of molecular mechanisms for suppression of MinE localization. These are not our main conclusion. Here we showed strong correlation between the suppression of MinE localization on the membrane and the wave generation, and the results do not rely on specific molecular mechanisms of the suppression. This is the reason why we argued these possible mechanisms in the Discussion. We agree that clarification of the molecular mechanism is an important future study.

For second mechanism, we should note that we mentioned “excluded volume effects” as an example that can causes the regulation of MinE’s conformational switching. In fact, we mentioned “excluded volume or other effects”.

The authors contradict both of these explanations with experiments or in the main text.

We disagree that our discussion is contradicted by the experimental results. This point is described in the following responses.

Re explanation 1): They performed the additional experiment requested in review, showing that neither labelled BSA nor sfGFP-MinD seem to bind to the membrane (Figure 2D). However, the setup of the experiment is problematic. The authors used 1 µM labelled BSA together with 1 mg/ml non-labeled BSA (about 16 µM). It could be that the signal of membrane-bound BSA is therefore too weak to be detected. To disentangle this, we would suggest the authors use different concentration of purely labelled BSA, starting with low concentrations.

To reply the concern by the reviewer, we tested 5 µM labelled BSA together with 5 µM non-labeled BSA (total 0.67 mg/mL). The results showed that localization of labelled BSA on membranes was not as similar to the case of 1 µM labelled BSA together with 1 mg/ml non-labeled BSA. The figure was replaced with the result using 5 µM labelled BSA together with 5 µM non-labeled BSA. Because 10 mg/mL BSA is a minimum concentration to emerge Min wave (Figure 3C), the results are not contradicted by the first mechanism. Moreover, we should mention that our experiments showed that sfGFP (not sfGFP-MinD) did not bind the membrane (Figure 2D).

Re explanation 2): The authors cite in the main text that BSA shows little depletion force compared to PEG8000 or Ficoll70. However, these polymer crowders do not support pattern formation, contradicting the statement that the depletion force is likely to be causing MinE conformational switching.

First, we did not mention that the depletion force is the only driving force of the mechanism. Although the depletion force is one of the effects derived from excluded volume effects, these are not identical. Second, we did not restrict ourselves to the cause of the secondmechanism in “excluded volume effects”. Hence, the logic is not contradictory to our experimental results. To clarify the point, we rephrased corresponding sentence as follows:

“… by excluded volume or other effects we do not consider.”

As reviewer #1 also pointed out, our description to deny the depletion force for the mechanism of the wave generation was not clear. We have revised the text.

The third explanation not mentioned by the authors could still be that BSA passivates the water oil interface, preventing MinE attachment and denaturation. This effect is hard to access experimentally, because it could also take place during droplet formation, when lipids and proteins come to the interface. If the lipid monolayer is not generated fast enough, proteins could denature, but a proper monolayer could still form. Hence, it would be more convincing if the FRAP data on the protein in presence and absence of BSA would not be presented for MinD, but for MinE.

To respond the reviewers’ concern, FRAP data on MinE in the absence of MinD was added in the new Figure 2—figure supplement 2. Because MinE localization changes by supplementation of BSA (Figure 2D), the result was obtained under no BSA conditions. FRAP data showed MinE moves faster than MinD. This result clearly shows that MinE does not denature during droplet formation even though BSA is absence. The following sentence is now added in the revised manuscript:

“Furthermore, FRAP also showed that MinE-mCherry diffuses faster than sfGFP-MinD in the absence of BSA, which indicates that the localization of MinE was not driven by protein denaturation (Figure 2—figure supplement 2).”

Furthermore, in the main text they demonstrate that I74M has a lower c/m ratio than WT MinE and is less influenced by BSA. They say that congruently this mutant does not support pattern formation. However, in the discussion they rightfully point out that this mutant does not show pattern formation under any of their conditions (the ones they tested) and also not in vivo. We still believe that this experiment needs to be removed in the current form, unless the authors can demonstrate that the mutant is able to form patterns in the droplet under some condition (e.g. at lower MinE concentration as demonstrated in vitro using the better characterized open MinE mutant I24N, see Denk et al., 2018) or by using the characterized MinE mutant I24N in the first place. The authors then need to show that the conditions for wave emergence are differentially influenced by BSA addition.

We appreciate the comment. After careful consideration, we decided to omit the I74M results and its discussion from the manuscript as the reviewer’s recommendation. The explanations of the open and closed states were moved to the Discussion.